# Effect of Using Glass Fiber Reinforced Polymer (GFRP) and Deformed Steel Bars on the Bonding Behavior of Lightweight Foamed Concrete

**Suhad M. Abd [1], Rafal Hadi [2], Shaker Abdal [3,\*], Saba Shamim [4], Hadee Mohammed Najm [5]** and **Mohanad Muayad Sabri Sabri [6]**

1. Department of Highways & Airports Engineering, College of Engineering, University of Diyala, Baqubah 32001, Iraq
2. Department of Civil Engineering, Bilad Alrafidain University College, Baghdad 32001, Iraq
3. Department of Civil Engineering, College of Engineering, University of Duhok, Duhok 42001, Iraq
4. Civil Engineering Section, University Polytechnic, Jamia Millia Islamia, New Delhi 110025, India
5. Department of Civil Engineering, Zakir Husain Engineering College, Aligarh Muslim University, Aligarh 202001, India
6. Peter the Great St. Petersburg Polytechnic University, 195251 St. Petersburg, Russia
* Correspondence: shaker.abdal01@gmail.com

**Abstract:** The study aims to conduct a direct pull-out test on fifty-four cube specimens considering different variables, including the type of reinforcement (sand-coated glass fiber-reinforced polymer (GFRP) and ribbed steel bars); the type of concrete (normal weight concrete NWC and lightweight foamed concrete LWFC); the diameter of the reinforcing bars (10 mm; 12 mm; and 16 mm) and the bonded length (3∅, 4∅, and 5∅). The hybrid fiber hooked-end steel (0.4% by volume) and polypropylene (0.2% by volume), respectively were used to improve the properties of LWFC by converting the brittle failure to ductile. The results showed that in the case of strengthened foamed concrete (FC), the bond strength with steel bars was greater compared to that with the GFRP bars. The bond strength ratio between the GFRP and steel bars of the FC specimens was found to vary between 37.8–89.3%. Additionally, in all specimens of FC, pull-out failure was witnessed with narrower crack width compared to NWC. Furthermore, mathematical equations have been proposed for predicting the bond strength of FC with steel and GFRP bars and showed good correlation with the experimental results.

**Keywords:** bonding behavior; ribbed steel bars; GFRP; foamed concrete; direct pull-out test; bond stress–slip relations

## 1. Introduction

Based on density, concrete has been classified into three categories, namely lightweight concrete (800–2000 kg/m$^3$), regular concrete (2000–2600 kg/m$^3$), and heavyweight concrete (>2600 kg/m$^3$) [1–3]. Foamed concrete is a type of lightweight concrete which mainly consists of the binding material, fine sand, and water; meanwhile, the air is filled by a homogeneous foam that is gradually added to the other components of the mixture until it reaches the required density. These homogeneous air bubbles replace the coarse aggregate in normal concrete, which is the fundamental difference between foamed concrete and normal concrete. Foamed concrete rather can be considered as a new building material, having limited application in the construction sector [4,5]. This may be attributed to the lack of research findings and data related to its structural properties, and bond behavior with the reinforcing bars.

By incorporating foamed concrete in the construction sector, exploitation of natural resources, mainly the coarse aggregates can be sustainably monitored [6,7]. Additionally, some other advantages offered by the foamed concrete are (i) enhanced thermal insulation,

consequently reducing the energy expenditure, (ii) consumption of harmful environmental wastes (such as silica fume, fly ash, and ground granulated blast-furnace slag (GGBS) [8,9]) with pozzolanic properties that can used as chemical binders, thereby reducing cement consumption and therefore high carbon emissions, (iii) ease in transportation of structural members due to its light weight, which reduces labor costs and therefore the cost of project [10].

The application of foamed concrete has become popular all over the world; one example of using foamed concrete as a construction material in Iraq is the city of residential architecture project in the Maysan region of Southern Iraq. Additionally, foamed concrete has been used for nonstructural applications (having a compressive strength of about 17 MPa) as a levelling material under tiles, instead of other waste materials.

Many research studies have been conducted to investigate the effect of adding different types and percentages of fibers on the mechanical properties of foamed concrete [11,12]. These findings have confirmed that fibers can significantly improve the properties of foamed concrete, especially polypropylene fibers and hooked-end steel fibers. Moreover, GFRP bars have gained popularity in the construction industry due to their high tensile strength, light weight and corrosion resistance which helps to maintain the life of buildings [13,14]. All these features enable them to be designed and fabricated in a number of fields, especially those with high load-carrying capacity, such as hydraulic engineering, building construction and highways.

The bond between the traditional concrete and reinforcement plays a crucial role in providing strength and durability to the structural members. Correspondingly, in the case of foamed concrete, it is essential to first evaluate the bond behavior of foamed concrete with reinforcing bars (herein, GFRP and steel bars were used). The bond behavior of lightweight concrete has been previously studied by a number of researchers. Nadir and Sujatha [15] studied the effect of bond strength by conducting a pull-out test on lightweight concrete containing coconut shell as coarse aggregate, with replacement levels of 25%, 50%, 75%, 100% and deformed steel bars with 12 mm and 16 mm diameters. A reduction in bond strength was reported with respect to the increase in coconut shell replacement percentage and the increase in the diameter of the bars, respectively. Zhao et al. [16] performed the pull-out test to investigate the bond strength of a lightweight aggregate containing expanded shale aggregates in its fine and coarse state with 100% replacement. A variety of variables, such as w/c ratio of the mix, bond length, and rebar diameter, were considered. The results indicated an increase in the bond strength upon decreasing the w/c ratio and upon increasing the diameter of the bars. Further, Abbas et al. [17] investigated the effect of different variables, such as bonded length, bar diameter, and concrete cover, on the bond strength of lightweight concrete (LWC) containing steel fibers and porcellanitic as coarse aggregate. It was found that the bond strength of normal weight concrete was higher compared to LWC [18,19]. As for the types of reinforcement, studies have proven that the use of deformed bars offers a higher resistance to bonds compared to the plain bars, significantly due to the presence of ribs which function as the rough surface and therefore enhance the frictional and mechanical forces, thus enhancing the bond [20–23]. Further, the bond behavior of glass FRP bars was studied by Tang et al. [24] after covering them with a layer of sand to improve the bond with the lightweight concrete. The bond resistance was found to be improved by 350% compared to the plain glass bars.

In contrast, the studies related to the bond behavior of foamed concrete are quite limited. Indeed, in international literature, only two studies on the bond behavior of foam concrete have been reported to date. Nindyawati and Umniati [25] performed the pull-out test to study the bond performance between bamboo-reinforced bars and nonstructural foamed concrete (12.7 MPa compressive strength), and reported the bond strength between 0.33–0.48 MPa. Additionally, in another research, de Villiers et al. [26] conducted two types of tests, namely beam end and pull-out tests to verify the bond strength between deformed steel bars and foamed concrete with different densities (1200 kg/m$^3$, 1400 kg/m$^3$, 1600 kg/m$^3$). A direct relationship between the density of foamed concrete and bond

strength was observed. Moreover, in the beam end test, early and sharp cracks were observed compared to normal concrete.

However, these studies are not sufficient to arrive at any conclusive research finding. An extensive investigation considering several other variables, such as different types of reinforcing bars, variation in diameter, etc., is further required to understand the bond behavior of structural foamed concrete. The present study aims to investigate the bonding behavior of foamed concrete and reinforcing bars using a pull-out test, according to RILEM [18,19,27–29]. Two types of reinforcing bars, namely ribbed steel bars and sand-coated GFRP bars, each having diameters (∅)of 10 mm, 12 mm and 16 mm and bonded length ratios of 3∅, 4∅ and 5∅, were used. Further, based on the experimental findings, mathematical equations have been proposed in this study for the prediction of the bond strength of the two types of bars with the structural lightweight foamed concrete (FC).

## 2. Experimental Program

### 2.1. Materials

Specimens of lightweight foamed concrete were prepared by using ordinary Portland cement and silica fume as binding material. Silica fume is softer than cement, with pozzolanic properties; it is considered a subproduct of the ferrosilicon mineral production process and primarily contains very fine amorphous particles of $SiO_2$. Its chemical analysis is shown in Table 1. In addition to the local river sand, silica sand (Figure 1) was added to be used as fine aggregate. A local synthetic foam agent was used to produce the foam. Moreover, chemical additive superplasticizers were used to reduce the w/c ratio and to obtain the required structural strength.

**Table 1.** Chemical analysis of silica fume and ordinary Portland cement.

| Chemical Composition | Silica Fume | Ordinary Portland Cement |
| :---: | :---: | :---: |
| | Results % | Results % |
| $Al_2O_3$ | 1.80 | 4.51 |
| $Fe_2O_3$ | 0.42 | 3.68 |
| CaO | 2.30 | 61.19 |
| MgO | 1.60 | 2.31 |
| $SiO_2$ | 90.56 | 21.44 |
| $Na_2O$ | 0.70 | 0.1 |
| $K_2O$ | 0.73 | 0.48 |
| $SO_3$ | 0.56 | 2.7 |
| Loss on ignition | 2.40 | 2.39 |

Hybrid fiber hooked-end steel (Hs) and polypropylene (Pp) were also added to prepare two types of concrete mix, while the normal concrete specimens were designed according to British standards [30], with a w/c ratio of 0.43, a compressive strength of 40 MPa, and a slump of 110 mm. The materials' properties are listed in Tables 2–8. The bond behavior of the lightweight foamed concrete was studied using two types of reinforcement bars: conventional ribbed reinforcing bars and sand-coated GFRP bars (see Tables 9 and 10).

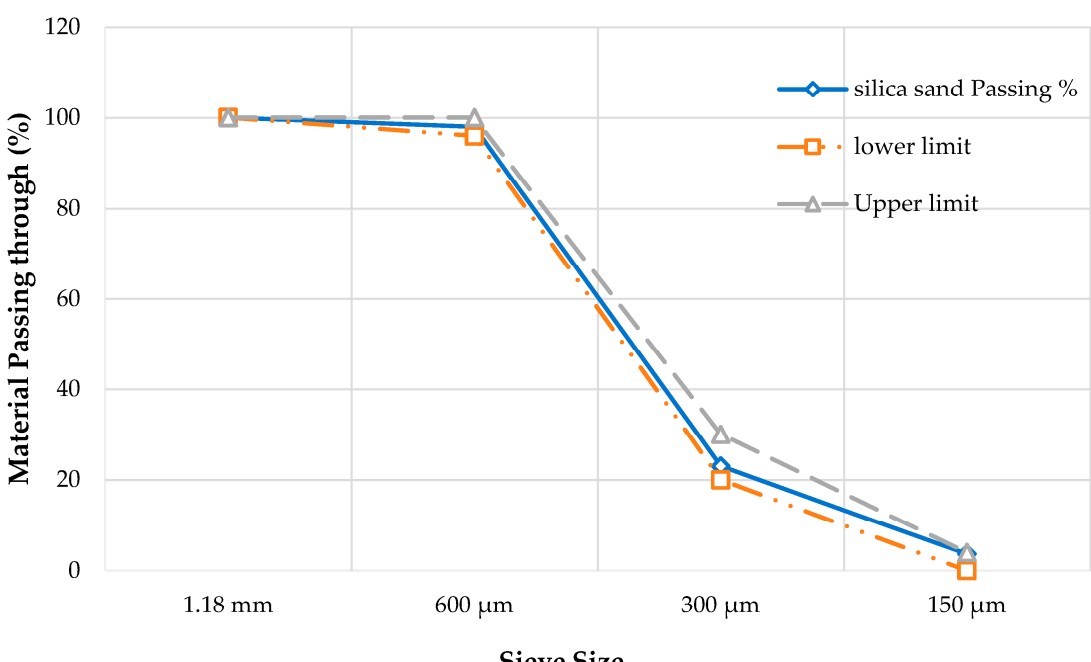

**Figure 1.** Grading of fine silica sand.

**Table 2.** Material properties of silica fume.

| Material | Color | Form | Specific Gravity | Bulk Density (kg/m$^3$) | Average Particle Size (μm) | Compressive Strength 7 Days (MPa) | pH Value |
|---|---|---|---|---|---|---|---|
| Silica fume | White | Powder | 2.6 ± 0.1 | 550–650 | 0.1 | 86.0 | 2 |

**Table 3.** Material properties of silica sand.

| Material | Specific Gravity | Bulk Density (kg/m$^3$) | Particle Size (mm) | Water Absorption (%) | Si$_2$O (%) | Porosity (%) |
|---|---|---|---|---|---|---|
| Silica sand | 2.6 | 1588 | 0.75–1.50 | 1.6 | 97 | 0.4 |

**Table 4.** Material properties of synthetic foam.

| Material | Color | Form | Freezing Point (°C) | Foam Expansion | Surface Tension | pH Value |
|---|---|---|---|---|---|---|
| Synthetic foam | Light yellow | Liquid | −5 to −30 | 4 ± 1% to 20% | 17.1 ± 10 % | 6.0–9.5 |

**Table 5.** Material properties of super plasticizer.

| Material | Color | Form | Component | Specific Gravity | Chloride Content | pH Value |
|---|---|---|---|---|---|---|
| Super plasticizer | Opaque | Liquid | Single | 1.06 ± 0.01 | Nil to BSEN 934-2 | 5.0–7.0 |

**Table 6.** Material properties of OPC.

| Material | Color | Fineness (m$^2$/kg) | Specific Gravity | Setting Time (Minutes) | | Compressive Strength (MPa) | | Water Absorption (%) | Si$_2$O (%) | Impact Value (%) | Crushing Value (%) |
|---|---|---|---|---|---|---|---|---|---|---|---|
| | | | | Initial | Final | 3 Days | 7 Days | | | | |
| OPC | Grey | 405 | 3.06 | 135 | 205 | 24.4 | 32.3 | 0.4 | 15.2 | 15.2 | 22.7 |

**Table 7.** Material properties of gravel.

| Material | Specific Gravity | Bulk Density (kg/m$^3$) | Maximum Particle Size (mm) |
|---|---|---|---|
| Gravel | 2.7 | 1734 | 12.5 |

**Table 8.** Material properties of fibers.

| Material | Bulk Density (kg/m$^3$) | Average Fiber Length (mm) | Average Fiber Diameter (mm) | Aspect Ratio L/D | Tensile Strength (MPa) | Ultimate Elongation (%) |
|---|---|---|---|---|---|---|
| Hs fiber | 7800 | 30 | 0.750 | 40.0 | >1100 | <2 |
| Pp fiber | 910 | 12 | 0.018 | 666.7 | 300–440 | - |

**Table 9.** Properties of steel bars.

| Diameters ⌀ (mm) | Surface Texture | Ultimate Stress $f_u$ (MPa) | Yield Stress $f_y$ (MPa) |
|---|---|---|---|
| 10 | Ribbed | 684 | 420 |
| 12 | Ribbed | 709 | 446 |
| 16 | Ribbed | 790 | 514 |

**Table 10.** Properties of GFRP bars.

| Diameters, ⌀ (mm) | Surface Texture | Tensile Strength (MPa) |
|---|---|---|
| 10 | Sand coated | 827 |
| 12 | Sand coated | 758 |
| 16 | Sand coated | 724 |

### 2.2. Mix Proportions

Normal weight concrete (NWC) specimens were designed according to British standards [30] to achieve a strength of 40 MPa, as shown in Table 11.

**Table 11.** Mix proportion of NWC.

| Cement (kg/m$^3$) | Sand (kg/m$^3$) | Gravel (kg/m$^3$) | Slump (mm) | w/c Ratio | Density kg/m$^3$ |
|---|---|---|---|---|---|
| 450 | 750 | 860 | 110 | 0.43 | 2320 |

Lightweight foamed concrete (LWFC) specimens were designed by conducting several trial mixes to study the effect of fiber addition in single (only one type of fiber, Hs or Pp) and hybrid (two types of fibers, Hs + Pp) states to obtain a lightweight concrete with a constant density of 1800 kg/m$^3$ and a compressive strength 40 MPa [31]. All mixing proportions are listed in Table 12. The schematic illustration of the mixing procedure of lightweight foamed concrete is shown in Figure 2, and the same has been summarized in the following paragraph.

1. The dry materials, cement, sand and silica fume, were circulated inside a 180 L rotary drum mixer for 30 s to evenly mix the dry materials with each other.
2. Then, 85% of the mixing water was added and mixed with the dry materials for approximately 2 min until homogeneous balls were obtained.
3. The superplasticizer (SP) was added to the remaining 15% of mixing water, and the latter was added to the previous mixture components and mixed for 1 min until the balls became a homogeneous mixture.
4. The foam was produced by the foam generator instrument and added to the mix immediately after preparation. The ingredients were mixed for at least 90 s until all foam was uniformly distributed and incorporated.

5. Finally, the fibers were added and spread over the mixture by hand, and then mixer was run for 30 s, ensuring its even distribution in the mix.

**Table 12.** Mixing proportions of LWFC.

| Mix Code | LWFC | LWFC Hs | LWFC Pp | LWFC Hs + Pp |
|---|---|---|---|---|
| Fibers (kg/m$^3$) | 0 | 31.20 | 1.83 | 31.20 + 1.83 |
| Cement (kg/m$^3$) | 1350 | 1350 | 1350 | 1350 |
| Sand (kg/m$^3$) | 900 | 900 | 900 | 900 |
| Silica fume (%) | 10 | 10 | 10 | 10 |
| Superplasticizer (SP) (%) | 1.12 | 1.12 | 1.12 | 1.12 |
| Water binder ratio | 0.28 | 0.28 | 0.28 | 0.28 |
| $f_c$ at 7 days (MPa) | 28.51 | 33.40 | 30.73 | 34.60 |
| $f_c$ at 28 days (MPa) | 34.00 | 38.06 | 37.12 | 40.80 |
| Foam (L/m$^3$) | 215 | 225 | 218.75 | 221.25 |

LWFC: Foamed concrete, LWFC Hs: Foamed concrete with hooked-end steel fibers, LWFC Pp: Foamed concrete with polypropylene fibers, LWFC Hs + Pp: Foamed concrete with hybrid fibers of hooked-end steel and polypropylene (FC), $f_c$: Cubic compressive strength.

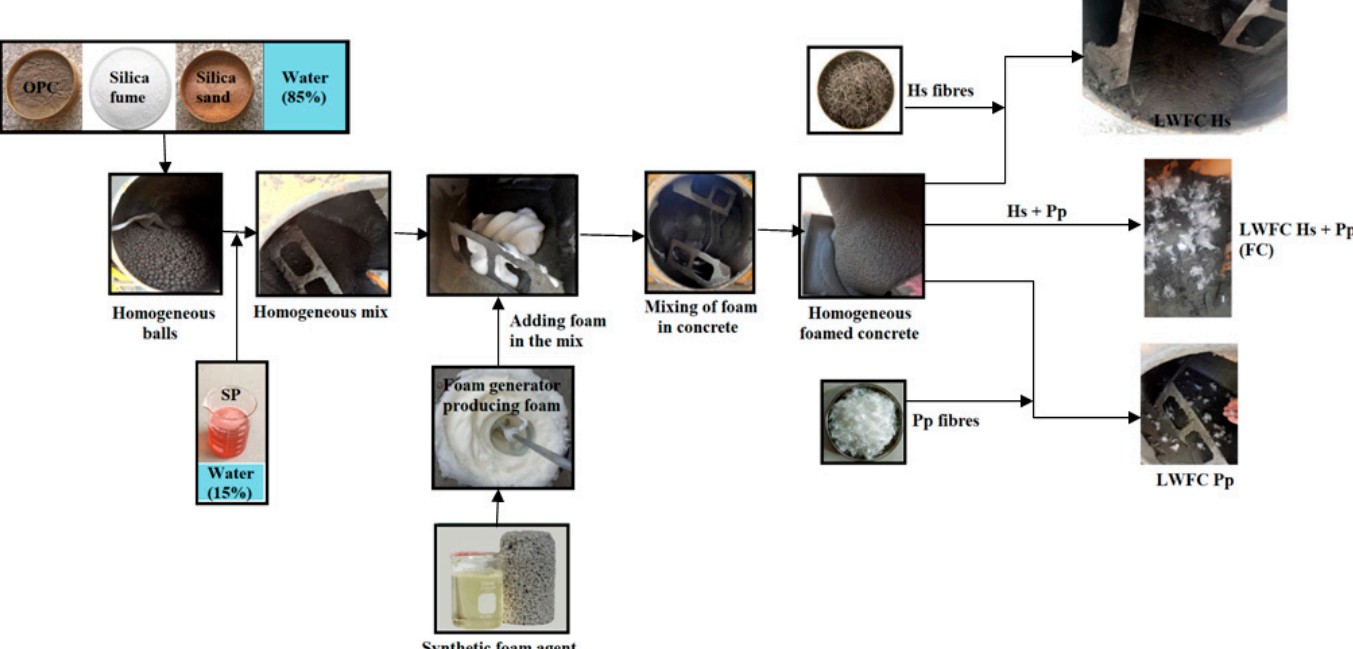

**Figure 2.** Preparation of mix.

For the present study, lightweight foamed concrete with hooked-end steel and polypropylene fibers, LWFC Hs + Pp (designated as FC) with a compressive strength of 40.8 MPa, was adopted to prepare the cube specimens for conducting the pull-out test.

*2.3. Mechanical Properties of Concrete*

After achieving the required design compressive strength of 40 MPa, other properties influencing the bond strength of concrete were also verified. A compressive strength test was carried out using cube specimens with dimensions of 100 × 100 × 100 (mm) [30]. A tensile strength test was conducted using cylinder specimens with dimensions of 100 × 200 (mm) [32]. A modulus of elasticity test was performed using cylinder specimens of size 150 × 300 (mm) [33], and a flexural strength test was conducted on prism specimens with dimensions of 140 × 140 × 600 (mm) [34].

### 2.4. Pull-Out Specimens

The pull-out test was conducted on a total of 54 cube specimens: 18 cube specimens of lightweight foamed concrete reinforced with sand-coated GFRP bars, 18 specimens of lightweight foamed concrete reinforced with traditional ribbed steel bars, and 18 specimens of normal weight concrete reinforced with traditional ribbed steel bars.

For this purpose, 18 cubic shaped wooden molds, each having a central circular opening at the base, were fabricated to cast the specimens. The diameter of the circular opening was kept equal to the diameter of the specific bar used in the test in order tightly confine the reinforcing bar through it, while the other end of the reinforcing bar was held by hand, ensuring that it was not tilted while pouring the concrete, as shown in Figure 3. The inner surface of the molds was coated with a thin layer of oil before pouring the concrete to facilitate easy removal of the specimen after hardening. Two types of reinforcing bars were adopted: steel bars and GFRP bars, each with diameters of 10 mm, 12 mm, and 16 mm, and bonded length ratios of $3\varnothing$, $4\varnothing$ and $5\varnothing$, which are equivalent to the following:

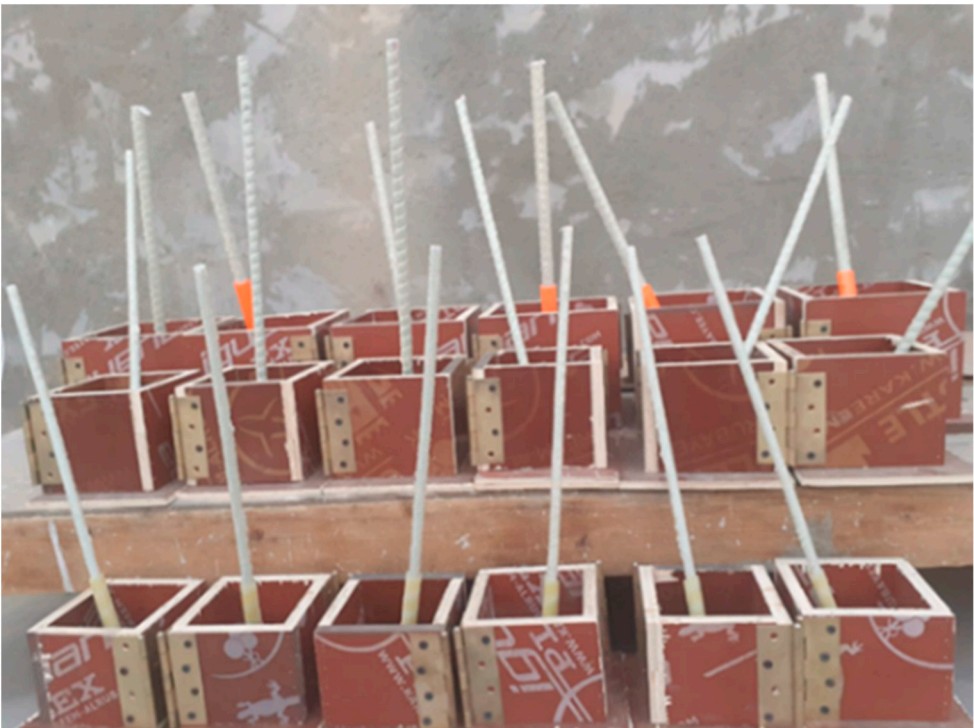

**Figure 3.** Prepared molds for casting the cube specimens.

30 mm, 40 mm, 50 mm for bars diameter 10 mm
36 mm, 48 mm, 60 mm for bars diameter 12 mm
48 mm, 64 mm, 80 mm for bars diameter 16 mm

Short bonded lengths were chosen because it difficult to obtain a uniform slip (displacement) along the bonding area while adopting a larger bond length [35,36], and this would eventually not fulfill the aim of the present study. The required bonding length between the concrete and the reinforcing bar was precisely determined, and the remaining portion which represents the unbonded length of the reinforcing bar was covered with a hollow plastic tube, as illustrated in Figure 4.

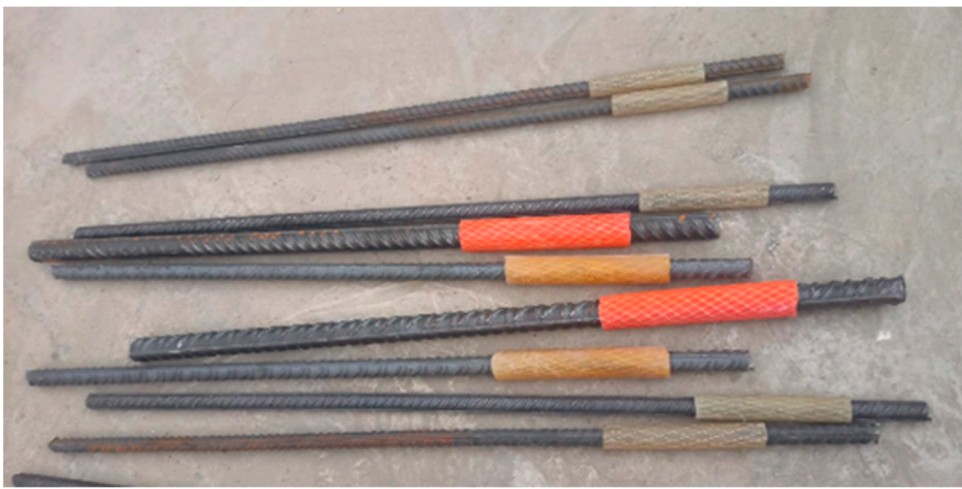

**Figure 4.** The unbonded lengths covered with plastic hollow tube.

After the molds were fully arranged, the mix of NWC was poured into molds and compacted in three layers, and the upper surface of concrete cubes were leveled and finished. In case of FC specimens, leveling, surface finishing and compaction were not required due to their self-compacting properties. After 24 h, the cube specimens were removed from the molds and cured for a period of 28 days. For both types of concrete (FC and NWC), the average of two specimens with the same variables was adopted to study the bond strength accurately.

*2.5. Specimens Code*

The code for all the specimens was derived in the following sequence of variables:

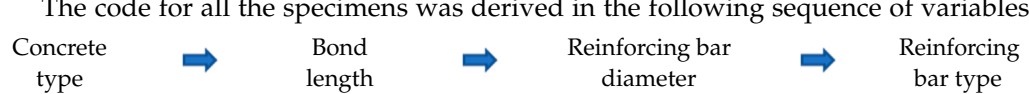

| Concrete type | Bond length | Reinforcing bar diameter | Reinforcing bar type |

Concrete (FC, NWC) Bond length (3∅, 4∅, 5∅) Bar diameter (10, 12, 16) Type (S—Steel).
Concrete (FC) Bond length (3∅, 4∅, 5∅) Bar diameter (10, 12, 16) Type (G—GFRP).

*2.6. Testing Machine and Setup*

The pull-out test was conducted using a hydraulic tensile testing machine (1000 kN capacity). The specimen was fixed tightly on the tensile testing machine using a steel frame consisting of two thick steel plates (25 mm thickness) in order to withstand the applied forces without any arcing or deviation during the testing process. The bottom plate had a central protrusion of certain length to enable proper fixing of the plate between the clamps of the test machine. The top plate had a central circular aperture for the reinforcing bar to pass through it. The two steel plates were tightly connected by heavy bolts after installing the specimen between them to confine the specimen during testing. The reinforced steel bar was fixed to the top clamp of the tensile testing machine. The specimens were subjected to compressive load (downward on the concrete cube) and tensile load (upward on the reinforcing bar) at a control displacement rate of 0.1 mm/s, as shown in Figure 5. The necessary data, tensile force and slip, were recorded during the experimental course directly from the testing machine using video imaging by the mobile camera until the test was completed.

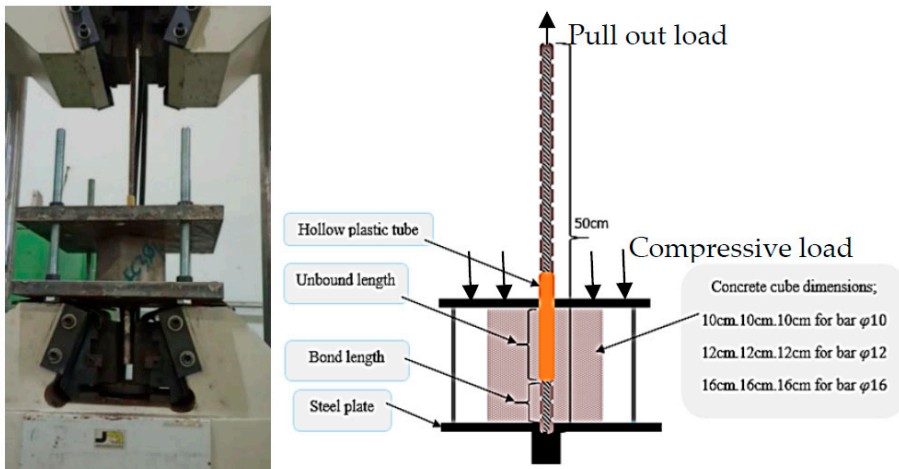

**Figure 5.** Schematic diagram of the pull-out test (**left**) and laboratory test setup on tensile testing machine (**right**).

In case of the FC specimens with GFRP reinforcing bars, some protection measures were taken on the loading end of the bars to avoid crushing due to the hydraulic pressure exerted during the pull-out test. It is an established fact that GFRP bars have high resistance to tensile load but offer weak resistance to compressive loading (here, clamping of bars for pull out test) and undergo brittle failure.

The protection involved covering the top portion of the GFRP bars with a standard sleeve tube stuffed with epoxy (Sikadur31). The sleeve tube used in the study was 130 mm in length, and its diameter was chosen based the minimum thickness of the epoxy around the GFRP bars but not less than 2.5 mm to avoid the slip between GFRP bar and the sleeve tube during tensile load application. Several trial tests on GFRP bars were conducted until the validity of the test procedure was verified satisfactorily without any slip between the GFRP bar and the sleeve tube, as shown in Figure 6. These protected GFRP bars were cured for 10 days for the epoxy to gain a desired compressive strength in the range of 50–60 MPa, as indicated in the datasheet provided by the company (Sika). A similar method for protection of the GFRP bars has been followed by Godat et al. [37] and Yang et al. [38].

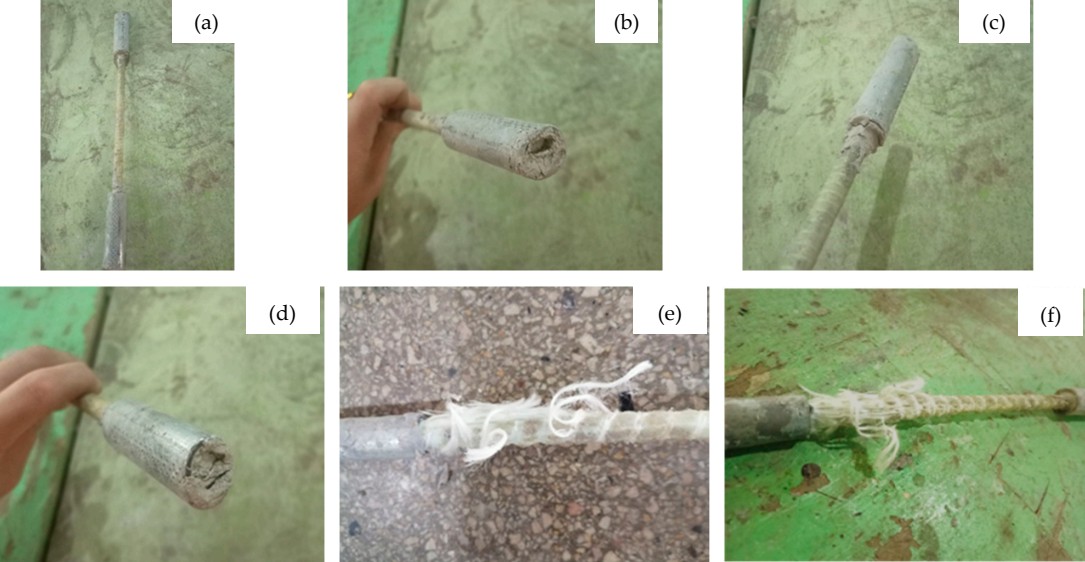

**Figure 6.** (**a**) Protected GFRP bar prepared for testing; (**b–d**) Failure of GFRP bar due to the bar slipping out of the sleeve tube (outside the scope of the study); (**e,f**) Yield failure of GFRP bar itself without any failure in the area of connection between the bars and the sleeve tube.

## 3. Results and Discussion

### 3.1. Mechanical Properties of Concrete

The results of the mechanical properties of NWC and LWFC have been summarized in Table 13.

**Table 13.** Results of mechanical properties of concrete.

| Mechanical Properties | $f'_c$ (MPa) | $f'_t$ (MPa) | $f'_r$ (MPa) | $E_c$ (GPa) | Slump (mm) | Flow (mm) | $\rho_d$ (kg/m$^3$) |
|---|---|---|---|---|---|---|---|
| NWC | 41.52 | 3.48 | 5.67 | 25.33 | 110 | - | 2325 |
| LWFC | 34.00 | 2.41 | 2.53 | 11.53 | - | 120 | 1820 |
| LWFC Hs + Pp (FC) | 40.80 | 4.59 | 5.83 | 24.60 | - | 100 | 1840 |

$f'_{cu}$: Cubic compressive strength, $E_c$: Modulus of elasticity, $f'_t$: Splitting tensile strength, $\rho_d$: Dry density, $f'_r$: Flexural strength (modulus of rupture).

The NWC mix was designed in accordance with the British standards, with compressive strength approximately equal to the compressive strength of FC. However, in the case of LWFC, the structural properties were improved by incorporating hooked-end steel fibers and polypropylene fibers into the mix to achieve a compressive strength of ~40 MPa. The compressive strength, splitting tensile strength, flexural strength and modulus of elasticity of FC were found to be enhanced by 20.0%, 90.8%, 130.6%, and 113.4%, respectively, compared to foamed concrete without fibers. This is because the fibers act as a connecting bridge inside the concrete which helps to restrict micro- and macro-cracks' formation [8,39,40]. In addition, fibers, especially PPF, bind with concrete via mechanical linking and interfacial interaction [41]. Moreover, these fibers restrict the propagation of shrinkage cracks during the hydration process, thus reducing the voids and greatly improving the strength of concrete [42], as shown in Figures 7–9. The flowability was within an acceptable range (115–140 mm) according to Hossain [43].

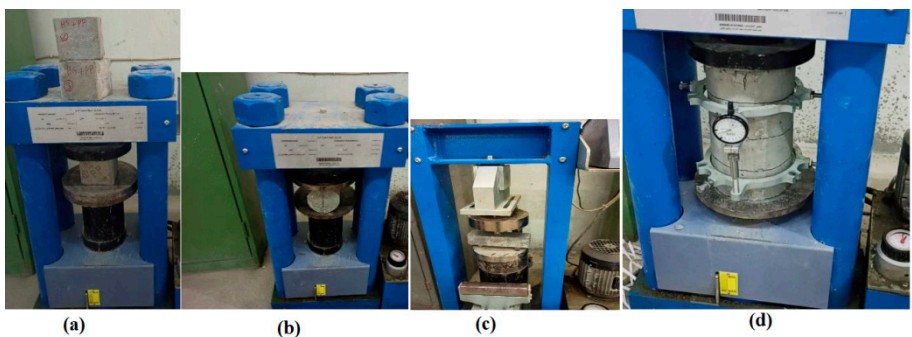

**Figure 7.** Foamed concrete specimen under (**a**) Compression test (**b**) Split tensile test (**c**) Flexural test (**d**) Modulus of elasticity test.

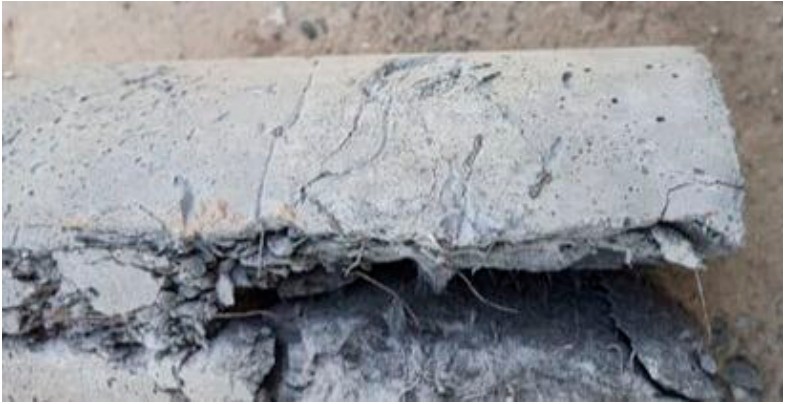

**Figure 8.** Splitting failure of FC.

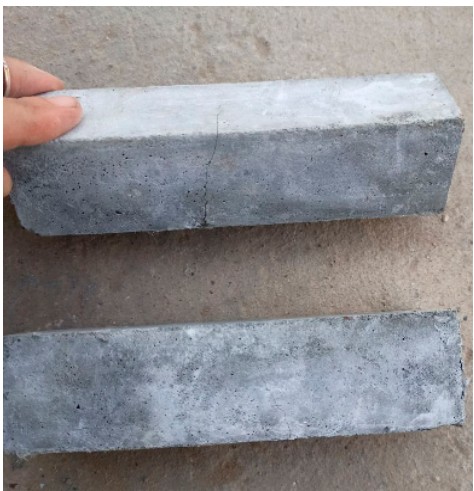

**Figure 9.** FC specimen after flexural test.

*3.2. Pull-Out Test Results*

In this section, the results for the bond strength have been presented (see Table 14) and were calculated using the following Equation (1);

$$\tau_u(\text{MPa}) = \frac{F}{\pi \varnothing l_b} \tag{1}$$

where

F is the maximum pull-out force (N),
$\varnothing$ is the rebar diameter (mm),
$l_b$ is the bonded length (mm).

It must be noted that in all the considered cases, the bond strength was calculated using average of two specimens of each type. Additionally, the longitudinal slip is defined as the displacement between the reinforcing bar and concrete at the loaded end, which was recorded directly from the testing machine using video imaging by the mobile camera until the test was completed.

**Table 14.** Results of pull-out tests.

| Specimen Code | Dimension (mm) | $S_u$ (mm) | $\tau_u$ (MPa) | $\frac{l_b}{\varnothing}$ | c (mm) | Mode of Failure |
|---|---|---|---|---|---|---|
| FC3$\varnothing$_10S | 100 × 100 × 100 | 4.3 | 24.66 | 3 | 45 | Pull-out |
| FC3$\varnothing$_12S | 120 × 120 × 120 | 4.6 | 22.24 | 3 | 54 | Pull-out |
| FC3$\varnothing$_16S | 160 × 160 × 160 | 5.3 | 18.91 | 3 | 72 | Pull-out |
| FC4$\varnothing$_10S | 100 × 100 × 100 | 3.8 | 27.59 | 4 | 45 | Pull-out |
| FC4$\varnothing$_12S | 120 × 120 × 120 | 8.0 | 24.39 | 4 | 54 | Pull-out |
| FC4$\varnothing$_16S | 160 × 160 × 160 | 8.1 | 21.00 | 4 | 72 | Pull-out |
| FC5$\varnothing$_10S | 100 × 100 × 100 | 5.1 | 17.28 | 5 | 45 | Pull-out |
| FC5$\varnothing$_12S | 120 × 120 × 120 | 7.8 | 20.52 | 5 | 54 | Pull-out |
| FC5$\varnothing$_16S | 160 × 160 × 160 | 6.0 | 17.11 | 5 | 72 | Pull-out |
| FC3$\varnothing$_10G | 100 × 100 × 100 | 10.2 | 18.97 | 3 | 45 | Pull-out |
| FC3$\varnothing$_12G | 120 × 120 × 120 | 4.1 | 15.03 | 3 | 54 | Pull-out |
| FC3$\varnothing$_16G | 160 × 160 × 160 | 7.3 | 7.14 | 3 | 72 | Pull-out |
| FC4$\varnothing$_10G | 100 × 100 × 100 | 9.7 | 20.86 | 4 | 45 | Pull-out |
| FC4$\varnothing$_12G | 120 × 120 × 120 | 2.5 | 15.62 | 4 | 54 | Pull-out |
| FC4$\varnothing$_16G | 160 × 160 × 160 | 5.6 | 10.13 | 4 | 72 | Pull-out |
| FC5$\varnothing$_10G | 100 × 100 × 100 | 4.1 | 12.00 | 5 | 45 | Pull-out |
| FC5$\varnothing$_12G | 120 × 120 × 120 | 8.2 | 18.33 | 5 | 54 | Pull-out |
| FC5$\varnothing$_16G | 160 × 160 × 160 | 8.0 | 6.73 | 5 | 72 | Pull-out |

**Table 14.** *Cont.*

| Specimen Code | Dimension (mm) | $S_u$ (mm) | $\tau_u$ (MPa) | $\frac{l_b}{\varnothing}$ | c (mm) | Mode of Failure |
|---|---|---|---|---|---|---|
| NWC3∅_10S | 100 × 100 × 100 | 6.8 | 24.83 | 3 | - | Pull-out |
| NWC3∅_12S | 120 × 120 × 120 | 5.0 | 19.35 | 3 | - | Pull-out |
| NWC3∅_16S | 160 × 160 × 160 | 3.1 | 19.17 | 3 | - | Pull-out |
| NWC4∅_10S | 100 × 100 × 100 | 5.0 | 28.65 | 4 | - | Splitting |
| NWC4∅_12S | 120 × 120 × 120 | 9.5 | 20.57 | 4 | - | Pull-out |
| NWC4∅_16S | 160 × 160 × 160 | 6.9 | 20.50 | 4 | - | Splitting |
| NWC5∅_10S | 100 × 100 × 100 | 5.1 | 19.11 | 5 | - | Splitting |
| NWC5∅_12S | 120 × 120 × 120 | 5.1 | 19.45 | 5 | - | Pull-out |
| NWC5∅_16S | 160 × 160 × 160 | 11.0 | 18.32 | 5 | - | Splitting |

$\tau_u$: Average ultimate bond strength, $S_u$: Average slip corresponding to the ultimate load, $l_b$: Bonded length, c: Concrete cover.

### 3.3. Bond Behavior of FC with Ribbed Steel Bar

#### 3.3.1. Effect of Bonded Length

Figures 10–12 depict a comparison of the bond stress–slip curve with varying bonded lengths (3∅, 4∅ and 5∅) for steel bars with a diameter of 10 mm, 12 mm and 16 mm, respectively, in FC. Corresponding to the bar of diameter 10 mm, the computed average ultimate bond strengths were found to be 24.66 MPa, 27.59 MPa and 17.28 MPa, respectively for bonded lengths 3∅, 4∅ and 5∅, respectively, while for the bar of diameter 12 mm, the same were computed as 22.24 MPa, 24.39 MPa and 20.52 MPa. For the bar of diameter 16 mm, these values were evaluated as 18.91 MPa, 21.0 MPa and 17.11 MPa, respectively. The bond strength has been found to be maximum for the bonded length 4∅, corresponding to all the considered diameters of the bars.

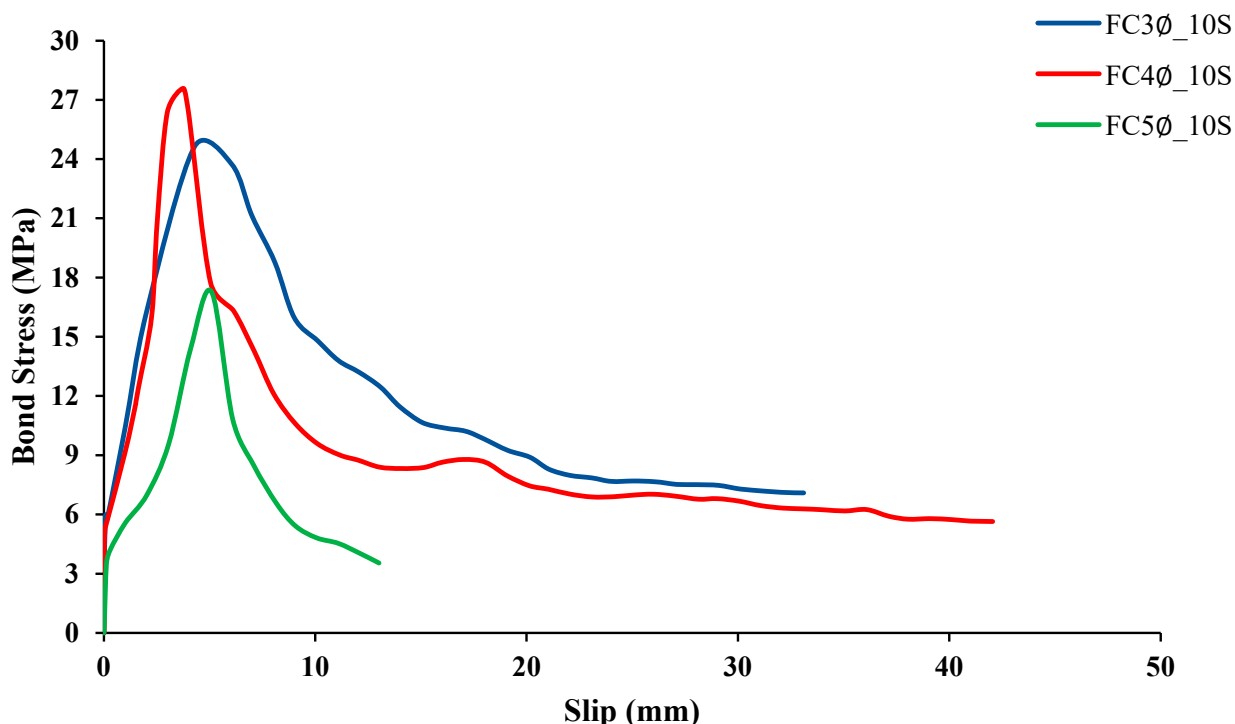

**Figure 10.** Effect of variation in bonded length for the 10 mm diameter steel bar.

For the bar of diameter 10 mm, the bond strength of FC4∅_10S was found to be 11.9% and 18.7% higher than FC3∅_10S ($l_b$ = 30 mm) and FC5∅_10S ($l_b$ = 50 mm), respectively. Usually, for large bonded lengths, the bond strength fails to show any enhancement due to the irregular distribution of stresses [16,44,45], which leads to a reduction in bond

stiffness along with the bond interface. Additionally, in Figure 10, the ascending branch and the descending branch of FC4∅_10S are found to be more converged compared to other bonding lengths due to the development of cracks on the surface of the FC4∅_10S specimen, as depicted in Figure 13d. This indicates a sudden reduction in bonding capacity compared to the continuous slow slippage of the bar from foamed concrete.

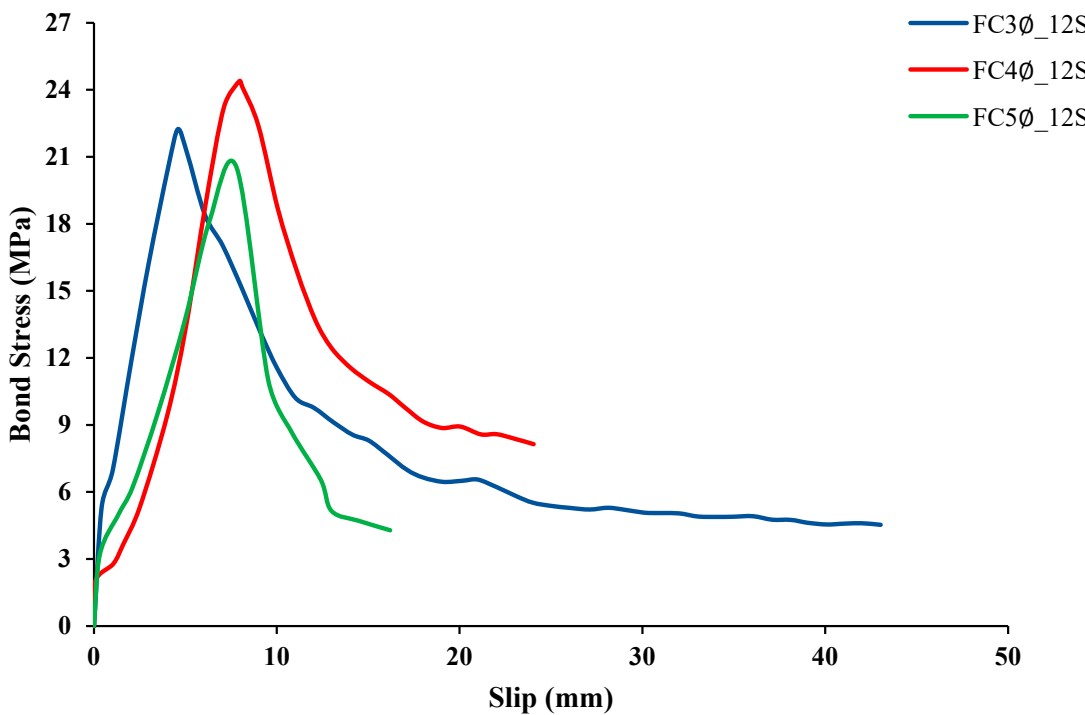

**Figure 11.** Effect of variation in bonded length for the 12 mm diameter steel bar.

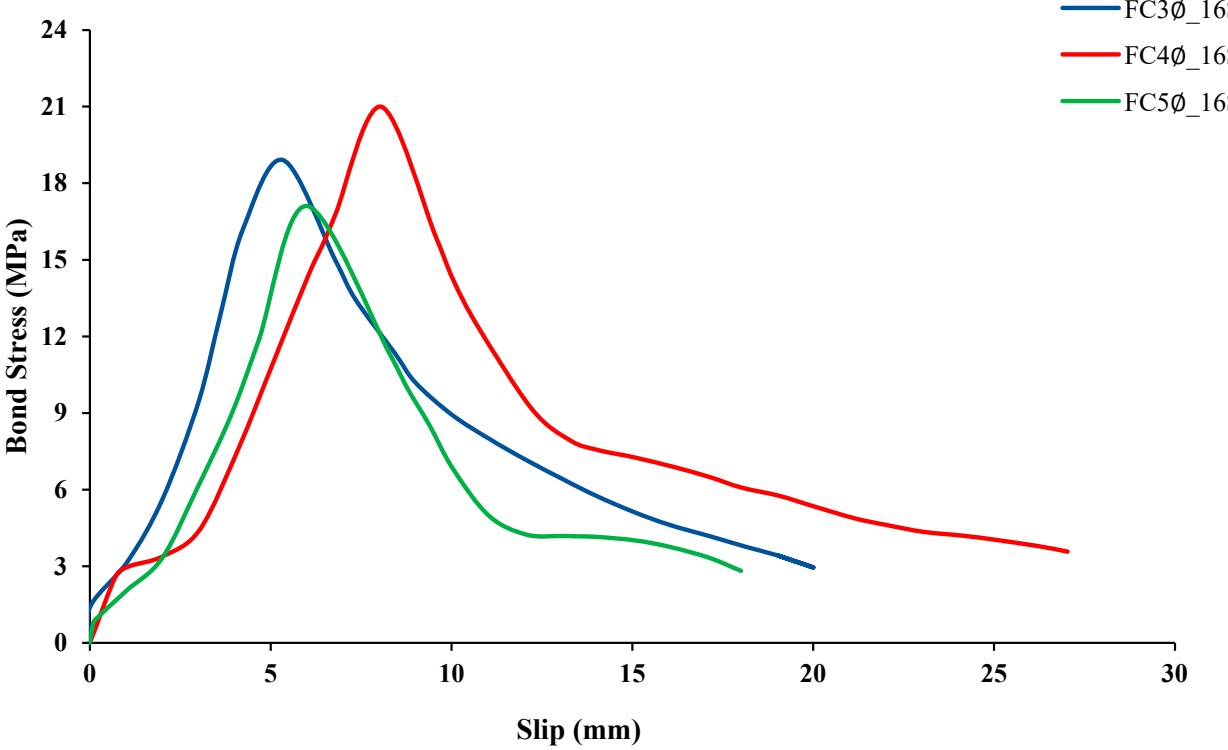

**Figure 12.** Effect of variation in bonded length for 16 mm diameter steel bar.

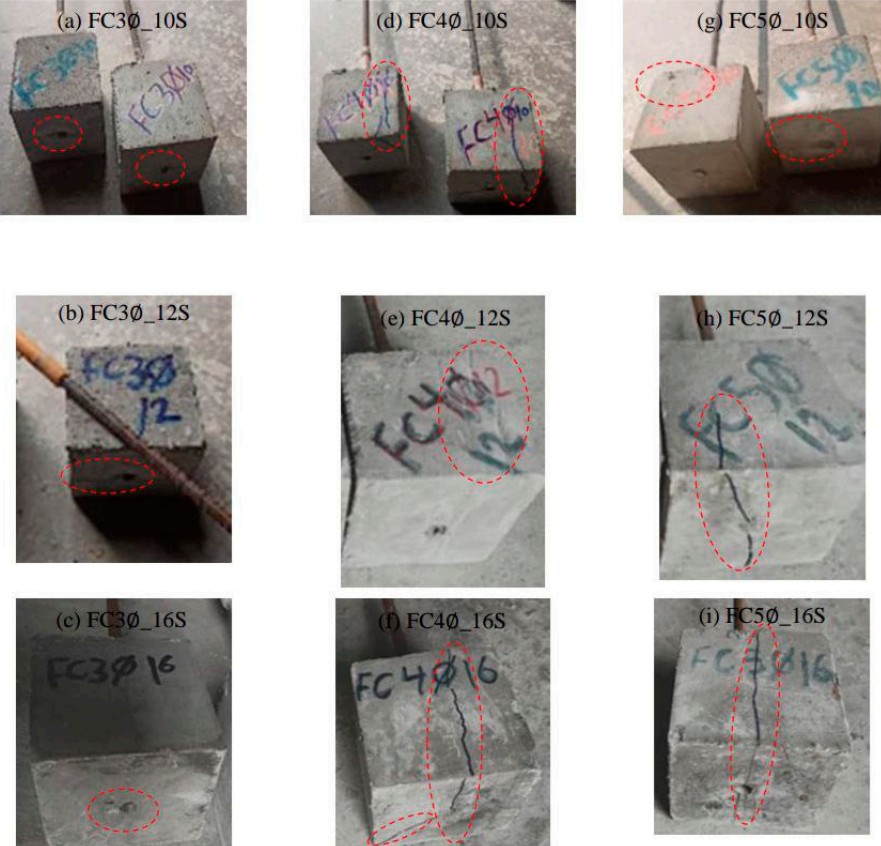

**Figure 13.** Failure patterns of FC reinforced with ribbed steel bars.

For the bar of diameter 12 mm, the bond strength of FC4∅_12S was found to be 9.7% and 59.7% greater compared to FC3∅_12S ($l_b$ = 36 mm) and FC54∅_12S ($l_b$ = 60 mm), respectively. Additionally, from Figure 11, it has been observed that the curve for FC5∅_12S is sharper compared to the other curves due to the formation of diagonal cracks around the reinforcing bar extending longitudinally to a specific height at the specimen interface, as shown in Figure 13h. These cracks appeared due to the crushing of concrete at the bonding interface as a result of high shear stresses between the ribs of the reinforcing bar and the surface of the concrete facing these ribs.

Furthermore, for the bar diameter of 16 mm, the bond strength of FC4∅_16S with a bonded length 64 mm was found to be 11.1% and 22.7% higher than FC3∅_16S ($l_b$ = 48 mm) and FC5∅_16S ($l_b$ = 80 mm), respectively, as shown in Figure 12. This is due to the rapid pulling out of the bar when the length of the bond is short, and the irregular distribution of stresses around the bars when the length of the bond is long. Figure 13c,i,f, illustrate the failure pattern; a longitudinal crack parallel to the steel bar was observed for the specimens FC3∅_16S and FC5∅_16S, while a diagonal crack around the reinforcing bar was witnessed in case of the specimen FC4∅_16S.

### 3.3.2. Effect of Bar Diameter

Figures 14–16 depict a comparison of bond stress–slip curves with varying diameters of steel bars (10 mm, 12 mm and 16 mm) for the bonded lengths 3∅, 4∅ and 5∅ in FC. Corresponding to bonded length 3∅, the computed average ultimate bond strengths were found to be 24.66 MPa, 22.24 Mpa and 18.91 Mpa, respectively, for bar diameters 10 mm, 12 mm and 16 mm; meanwhile, for bonded length 4∅, the same were computed as 27.59 MPa, 24.39 MPa and 21.00 MPa. For bonded length 5∅, these were evaluated as 17.28 MPa, 20.52 MPa and 17.11 MPa, respectively. Therefore, in case of the shorter bonded length (3∅ and 4∅), the bond strength has been found to have an inverse relation with the

diameter of the bar. In contrast, for the longer bonded length (5∅), the relation no longer obeyed the inverse proportionality. The bond strength was found to be maximum for the 12 mm diameter bar corresponding to bonded length 5∅. However, the bond strength was enhanced by 30.4% and 31.4% when the diameter from was reduced from 16 mm to 10 mm, corresponding to shorter bonded lengths 3∅ and 4∅, respectively. The results stipulate that FC offers weak confinement to the bars with larger diameters when shorter bond lengths are considered.

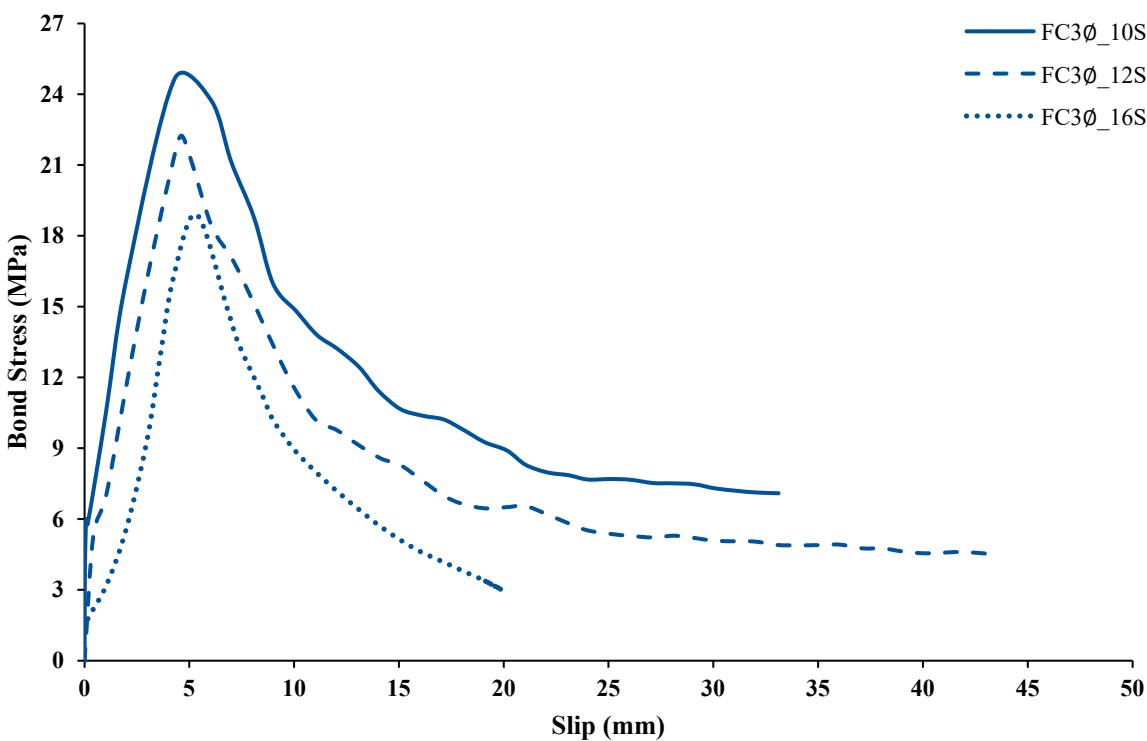

**Figure 14.** Effect of variation in steel bar diameter for bonded length 3∅.

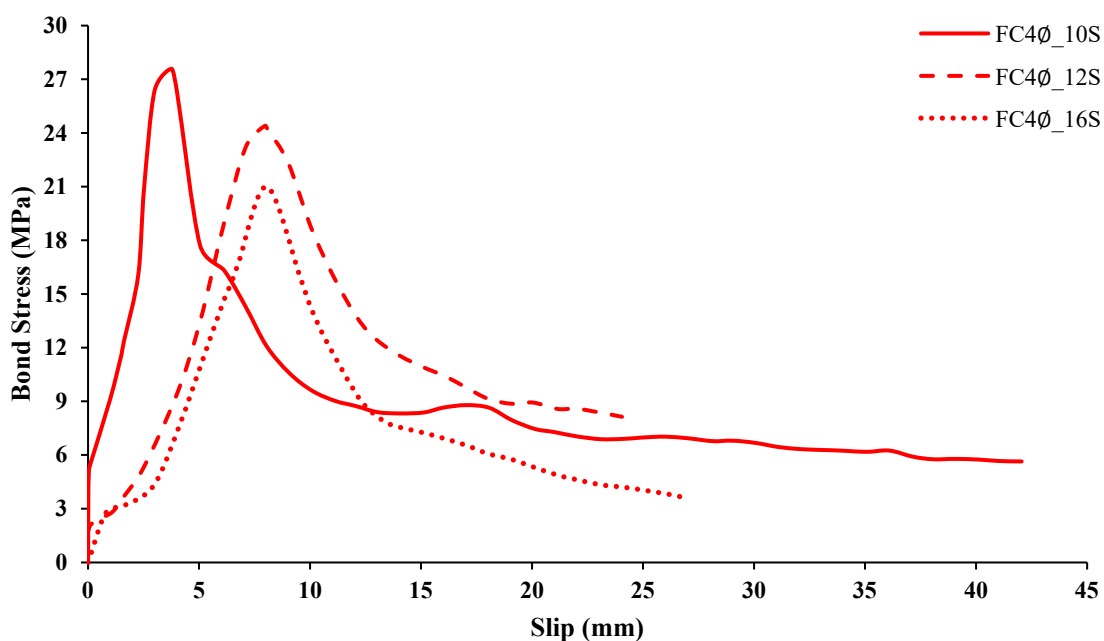

**Figure 15.** Effect of variation in steel bar diameter for bonded length 4∅.

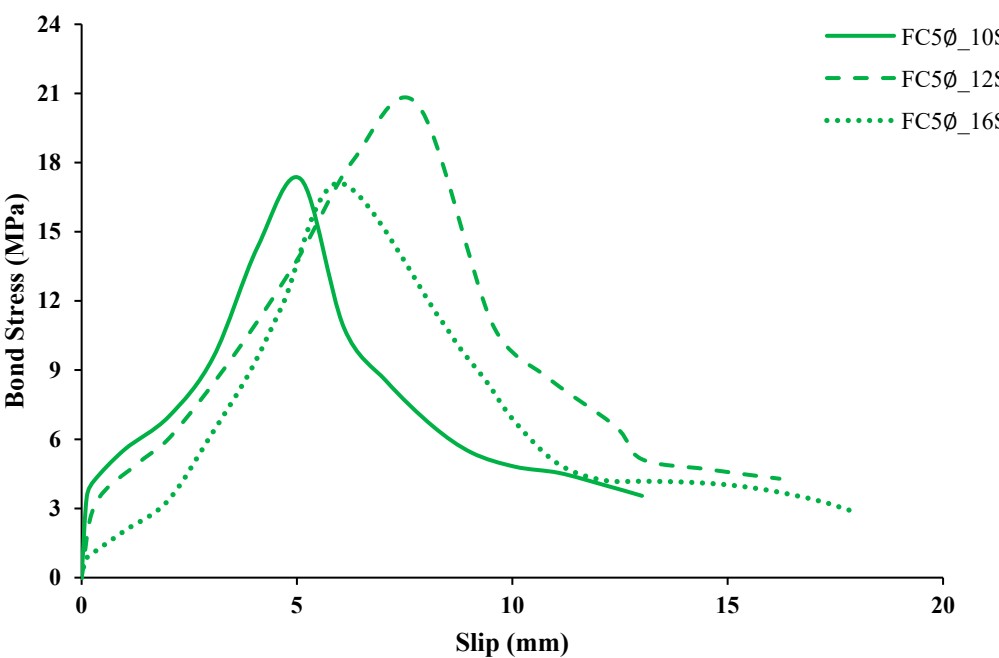

**Figure 16.** Effect of variation in steel bar diameter for bonded length 5∅.

*3.4. Bond Behavior of FC with Sand-Coated GFRP Bars*

3.4.1. Effect of Bonded Length

Figures 17–19 depict a comparison of bond stress–slip curves with varying bonded lengths (3∅, 4∅ and 5∅) for sand-coated GFRP bars with a diameter of 10 mm, 12 mm and 16 mm, respectively, in FC. Corresponding to the bar of diameter 10 mm, the computed average ultimate bond strengths were found to be 18.97 MPa, 20.86 MPa and 12.00 MPa for bonded lengths 3∅, 4∅ and 5∅, respectively, while for the bar of diameter 12 mm, the same were computed as 15.03 MPa, 15.62 MPa and 18.33 MPa. For the bar of diameter 16 mm, these were found to be 7.14 MPa, 10.13 MPa and 6.73 MPa, respectively. Similarly, to the steel bars, the bond strength of GFRP bars has been found to be maximum for the bonded length 4∅, corresponding to the 10 mm and 16 mm diameter bars, while for the 12 mm diameter bar, the bond strength was found to be highest for bonded length 5∅.

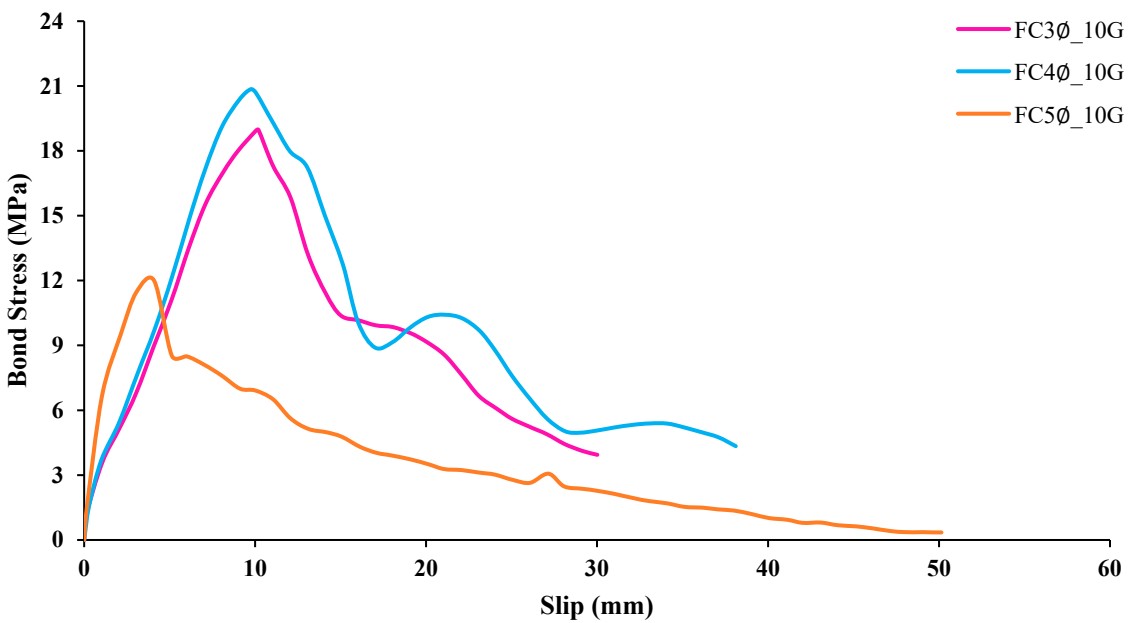

**Figure 17.** Effect of variation in bonded length for the 10 mm diameter sand-coated GFRP bar.

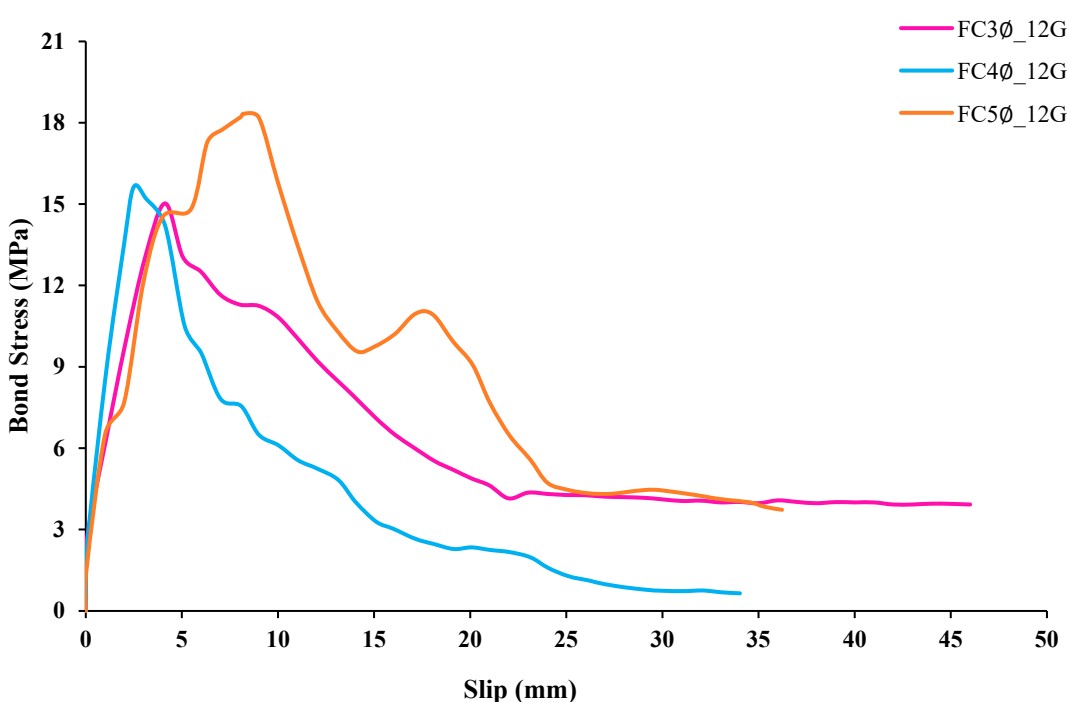

**Figure 18.** Effect of variation in bonded length for the 12 mm diameter GFRP bar.

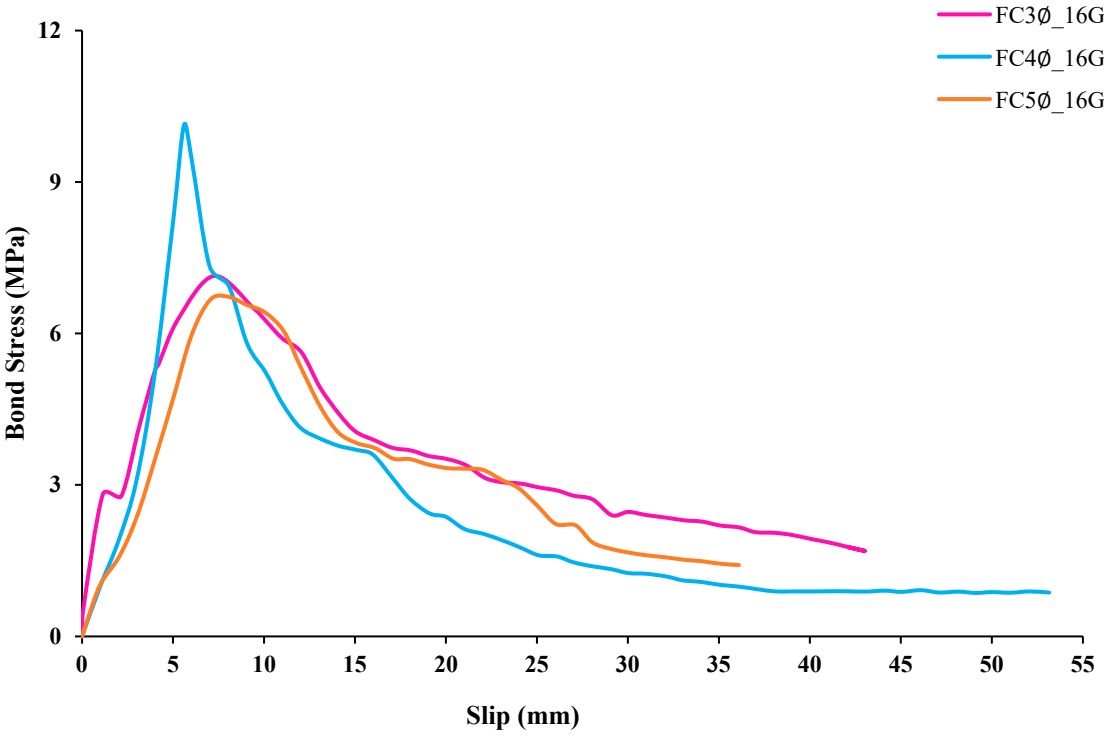

**Figure 19.** Effect of variation in bonded length for 16 mm diameter GFRP bar.

For the bar of diameter 10 mm, the bond strength of FC4∅_10G was found to be 10.0% and 30.2% higher than FC3∅_10G ($l_b$ = 30 mm) and FC5∅_10G ($l_b$ = 50 mm) bonded lengths, respectively. Again, for large bonded lengths, the bond strength fails to show any enhancement due to the irregular distribution of stresses [16,44,45] leading to a reduction in bond stiffness along with the bond interface.

For the bar of diameter 12 mm, the bond strength of FC4∅_12G ($l_b$ = 48 mm) was found to be 4.0% greater compared to FC3∅_12G ($l_b$ = 36 mm) and 13.8% lesser compared to FC54∅_12G ($l_b$ = 60 mm).

Additionally, for the bar of diameter 16 mm, the bond strength of FC4∅_16G with $l_b$ = 64 mm was found to be 41.9% and 50.5% higher than FC3∅_16G ($l_b$ = 48 mm) and FC5∅_16G ($l_b$ = 80 mm) bonded lengths, respectively, as shown in Figure 19.

In Figures 17–19, all the curves begin with a steady rise, up to the maximum bond stress. After that, the pull-out failure occurs, wherein the slip begins to increase rapidly and is accompanied by a rapid decrease in the bond capacity, indicated by the descending branch of the curve. The failure pattern of these specimens is depicted in Figure 20, in which only a few micro cracks are visible.

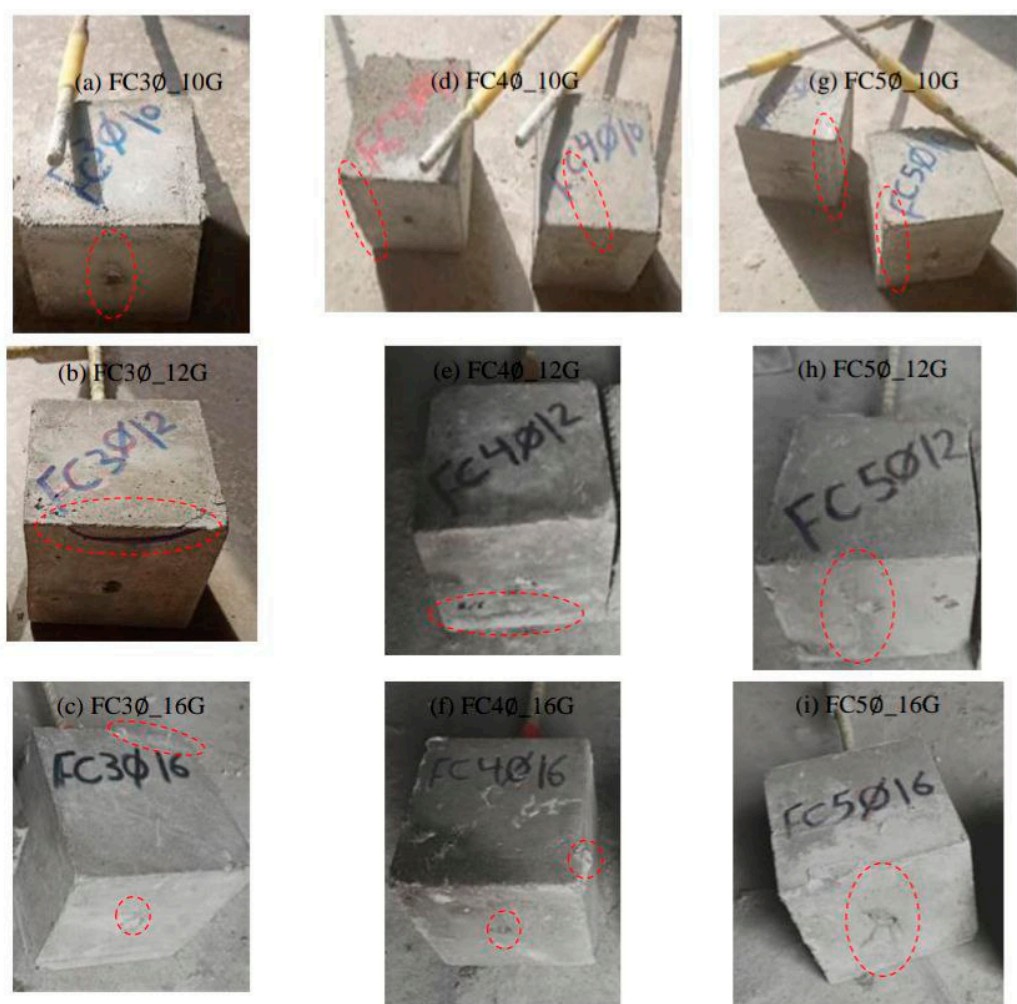

**Figure 20.** Failure patterns of FC reinforced with sand- coated GFRP bars.

### 3.4.2. Effect of Bar Diameter

Figures 21–23 depict a comparison of the bond stress–slip curves with varying diameters of sand-coated GFRP bars (10 mm, 12 mm and 16 mm), for the bonded lengths of 3∅, 4∅ and 5∅, in FC. Corresponding to bonded length 3∅, the computed average ultimate bond strengths were found to be 18.97 MPa, 15.03 MPa and 7.14 MPa, respectively for bars of diameter 10 mm, 12 mm and 16 mm; meanwhile, for bonded length 4∅, the same were computed as 20.86 MPa, 15.62 MPa and 10.13 MPa, and for bonded length 5∅, these were 12 MPa, 18.33 MPa and 6.73 MPa, respectively. Therefore, in case of GFRP bars, for a constant bonded length, an increase in the bond stress has been witnessed upon decreasing the diameter of the bars for specimens with shorter bonded lengths (3∅ and

4∅); this is similar to the case of steel bars. The bond strength was enhanced by 165.7% and 105.9%, corresponding to bonded lengths of 3∅ and 4∅, respectively, upon reduction of the diameter from 16 mm to 10 mm. On the other hand, for the longer bonded length (5∅), the bond strength was found to be maximum in the 12 mm diameter bar.

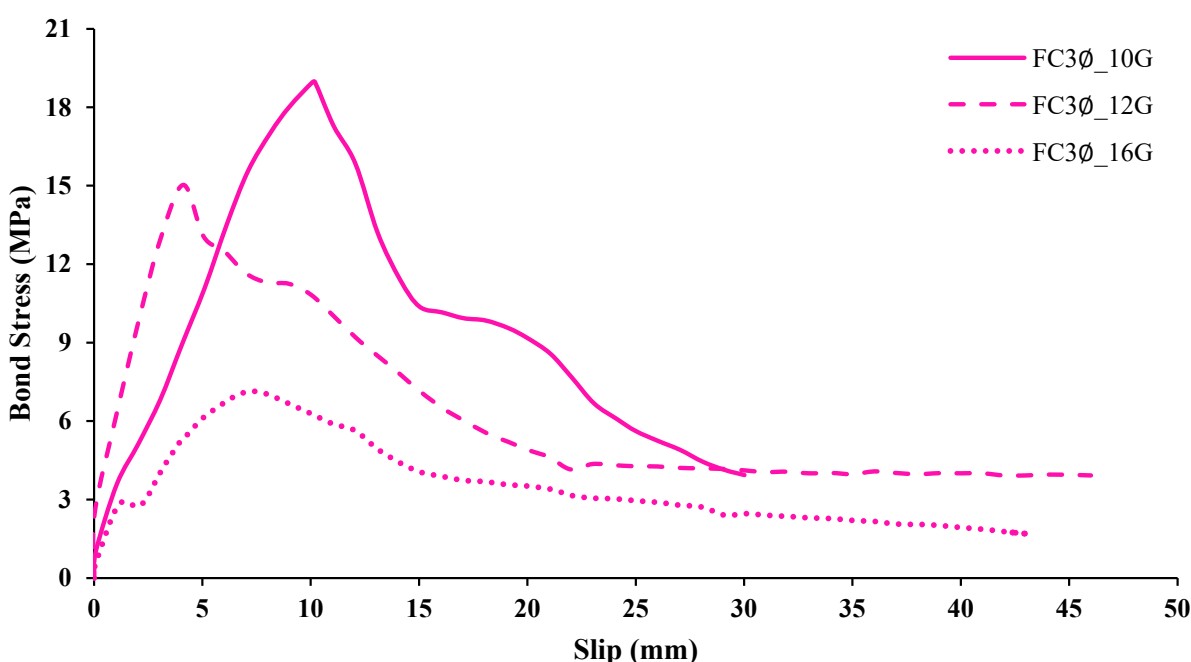

**Figure 21.** Effect of variation in GFRP bars' diameter for bonded length 3∅.

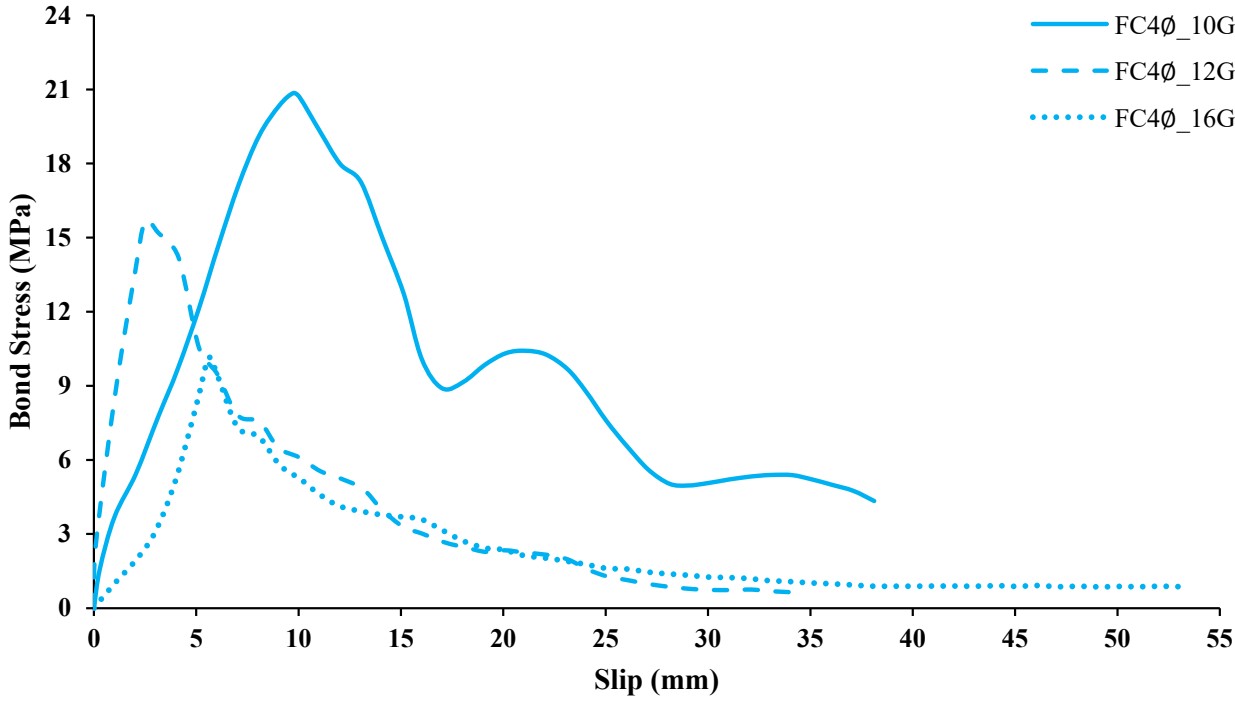

**Figure 22.** Effect of variation in GFRP bars' diameter for bonded length 4∅.

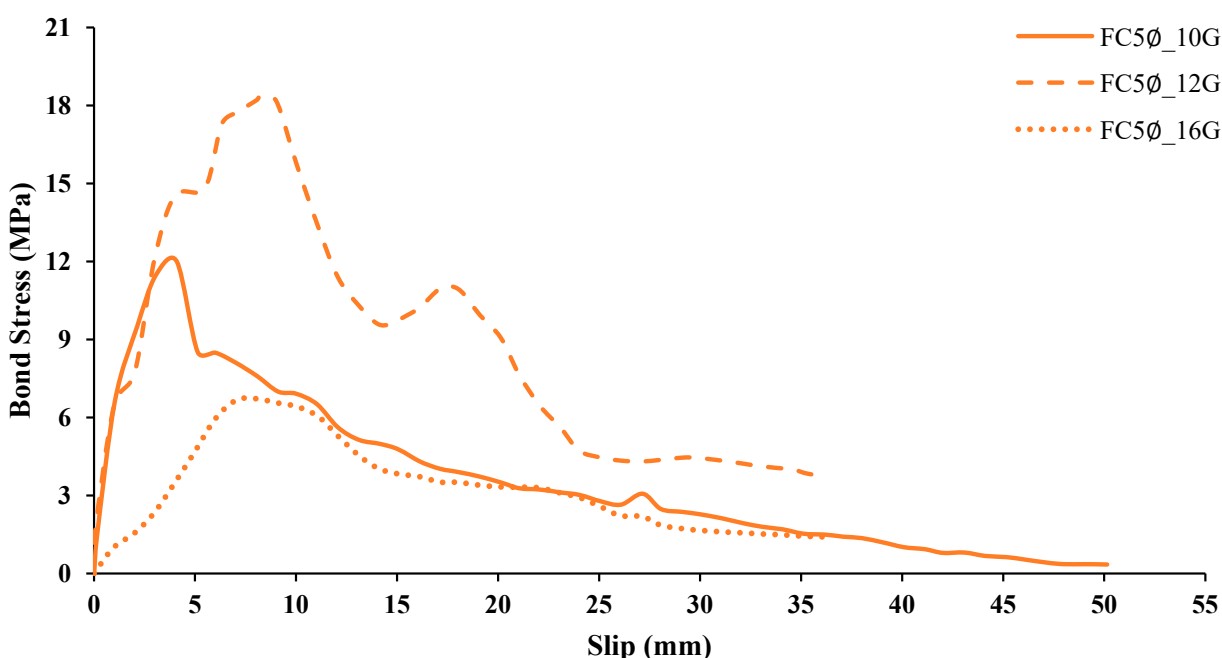

**Figure 23.** Effect of variation in GFRP bars' diameter for bonded length 5∅.

*3.5. Bond Behavior of NWC with Ribbed Steel Bar*

3.5.1. Effect of Bonded Length

Figures 24–26 depict a comparison of bond stress–slip curve with varying bonded lengths (3∅, 4∅ and 5∅) for steel bars with a diameter of 10 mm, 12 mm and 16 mm, respectively, in NWC.

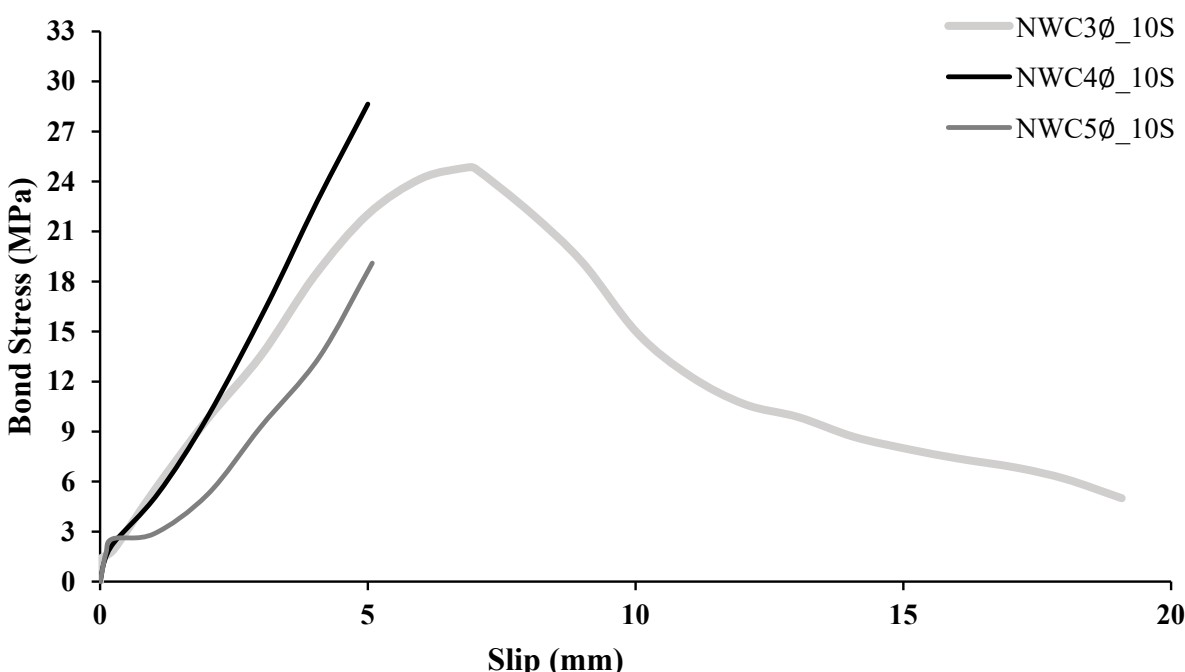

**Figure 24.** Effect of variation in bonded length for the 10 mm diameter steel bar.

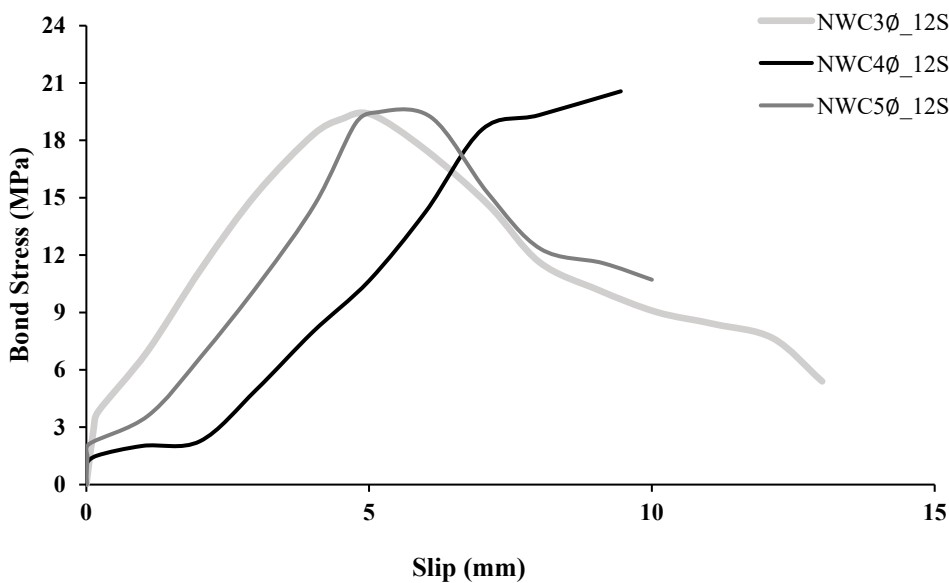

**Figure 25.** Effect of variation in bonded length for the 12 mm diameter steel bar.

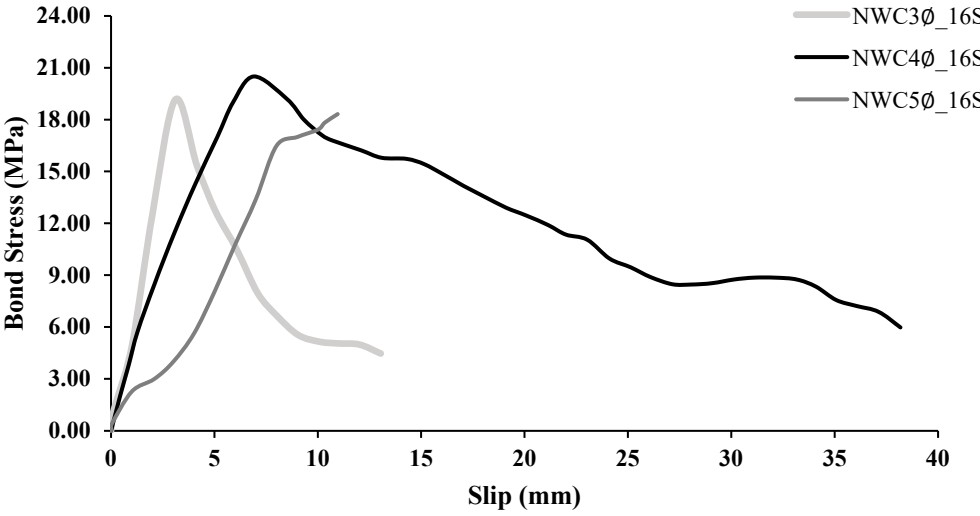

**Figure 26.** Effect of variation in bonded length for the 16 mm diameter steel bar.

Corresponding to the bar with a diameter of 10 mm, the computed average ultimate bond strengths were found to be 24.83 MPa, 28.65 MPa and 19.11 MPa, respectively for bonded lengths 3∅, 4∅ and 5∅, respectively, while for the bar of diameter 12 mm, the same were computed as 19.35 MPa, 20.57 MPa and 19.45 MPa, and for the bar of diameter 16 mm, these were 19.17 MPa, 20.5 MPa and 18.32 MPa, respectively. The bond strength has been found to be maximum for the bonded length 4∅, corresponding to all the considered diameters of the bars.

Therefore, for the bar of diameter 10 mm, the bond strength of FC4∅_10S was found to be 15.4% and 50.0% higher than FC3∅_10S ($l_b$ = 30 mm) and FC5∅_10S ($l_b$ = 50 mm) bonded lengths, respectively. In Figure 24, the ascending and the descending branch were recorded only for bonded length 3∅, while for 4∅ and 5∅, only the assending branch was observed due to the sudden shear failure of the specimen.

For the bar of diameter 12 mm, the bond strength of FC4∅_12S was found to be 6.3% and 5.8% greater compared to FC3∅_12S ($l_b$ = 36 mm) and FC54∅_12S ($l_b$ = 60 mm), respectively. Additionally, in Figure 25, the descending branch for 4∅ was not observed due to the sudden shear failure of specimen.

Furthermore, for the bar of diameter 16 mm, the bond strength of FC4∅_16S with a bonded length of 64 mm was found to be 6.9% and 11.9% higher than FC3∅_16S ($l_b$ = 48 mm) and FC5∅_16S ($l_b$ = 80 mm), respectively, as shown in Figure 26. The failure patterns of these specimens are depicted in Figure 27.

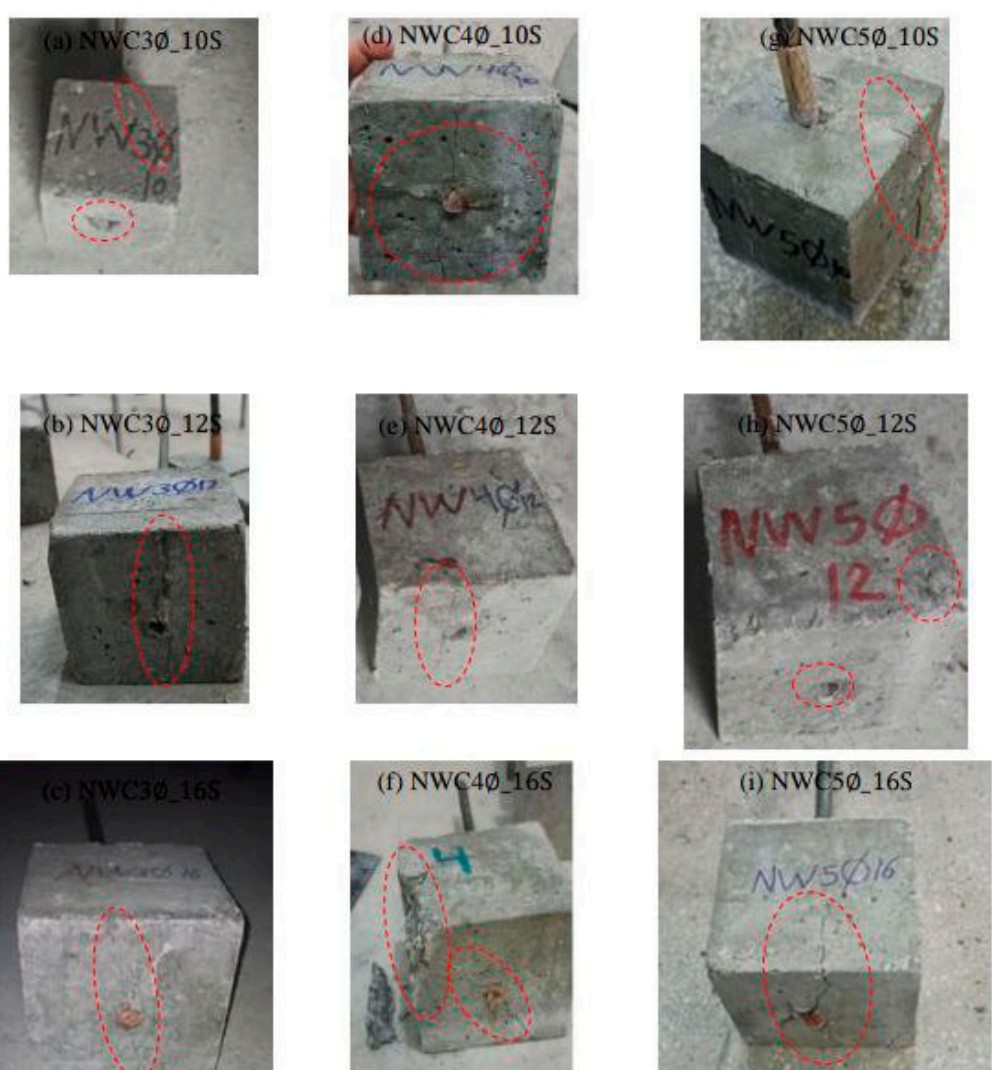

**Figure 27.** Failure patterns of NWC reinforced with ribbed steel bars.

### 3.5.2. Effect of Bar Diameter

A comparison of bond stress–slip curves with varying diameters of steel bars (10 mm, 12 mm and 16 mm) for the bonded lengths—3∅, 4∅ and 5∅, respectively, in NWC is shown in Figures 28–30. A similar analogy has been carried out in the case of NWC and FC with steel bars, i.e., for a constant bonded length, an increase in the bond strength has been observed upon decreasing the diameter of the bars with a shorter bonded length (3∅). The bond strength was enhanced by 26.1%, corresponding to the bonded length 3∅, upon reducing the diameter from 16 mm to 10 mm. On the other hand, for bonded length 4∅ and 5∅, proper peaks of bond stress could not be determined due to the sudden failure of the specimens.

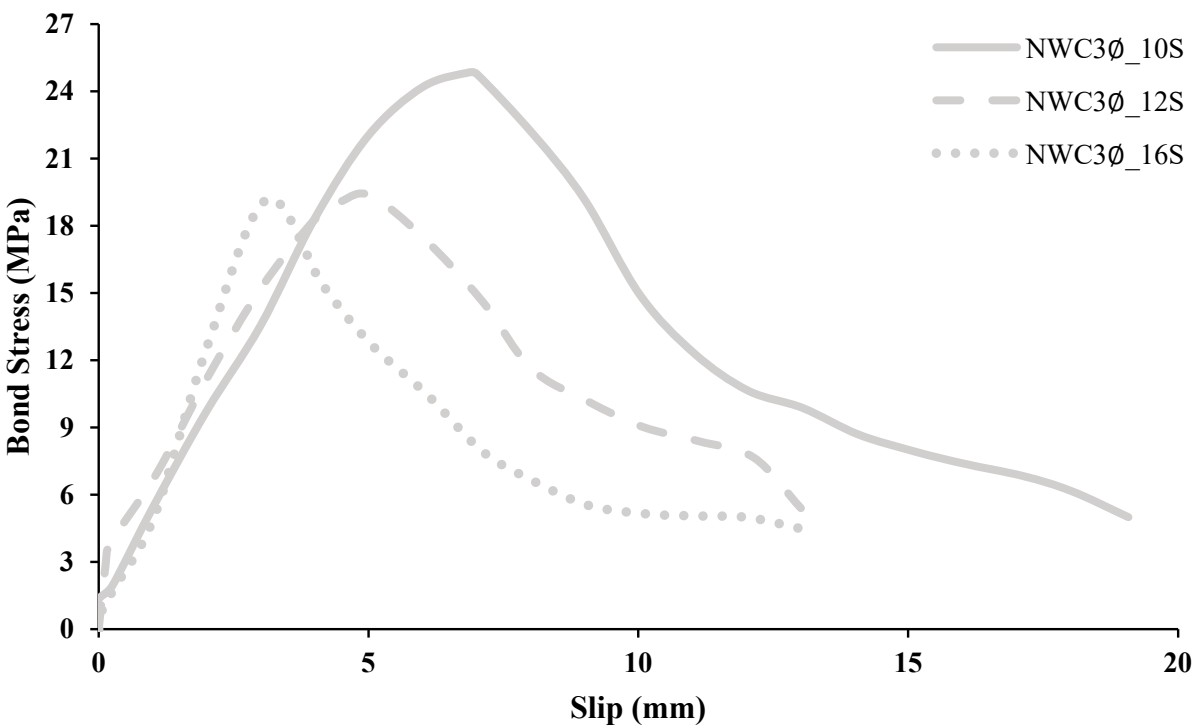

**Figure 28.** Effect of variation in steel bars' diameter for bonded length 3∅.

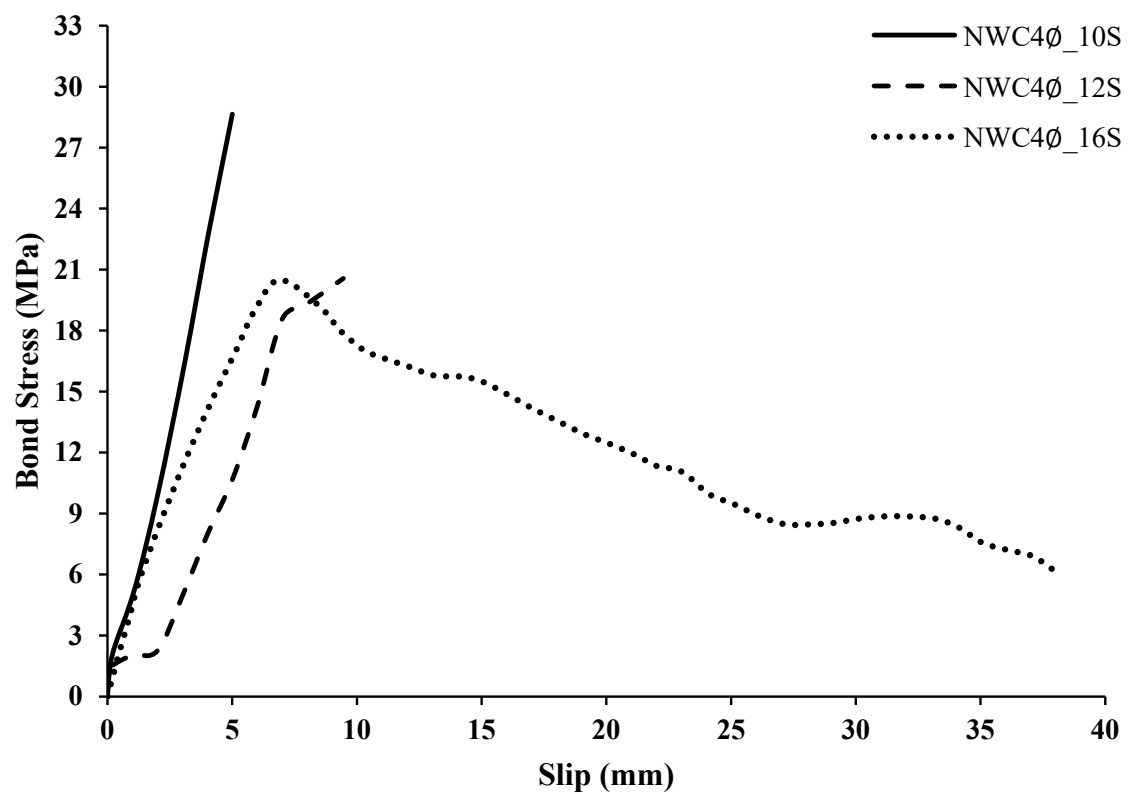

**Figure 29.** Effect of variation in steel bars' diameter for bonded length 4∅.

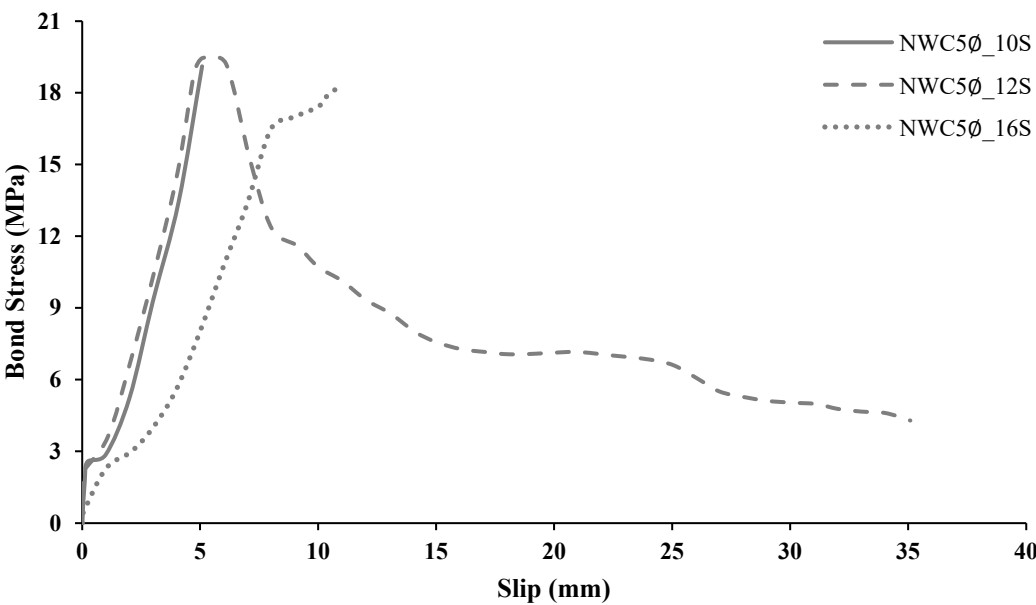

**Figure 30.** Effect of variation in steel bars' diameter for bonded length 5∅.

### 3.6. Effect of Type of Reinforcing Bar

Table 15 compares the bond strength between the two groups of foamed concrete with the same variables except for the type of reinforcing bar (Group 1: FC + Steel bar and Group 2: FC + GFRP bar). It has been clearly observed that bond strength in the case of steel bars is greater than that of the GFRP bars in FC specimen; this can be attributed to the high modulus of elasticity, increased chemical adhesion and greater surface roughness of steel bars compared to the GFRP bars. A similar variation in bond strength between steel and GFRP bars has been reported by Munoz [46]. The bond strength ratio between GFRP and steel bars of FC specimens $\left(\tau_{u,FC}^{G}/\tau_{u,FC}^{S}\right)$ is found to vary between 37.8–89.3%.

**Table 15.** Comparison of pull-out test results for the two types of reinforcing bars.

| Specimen Code | $\tau_{u,FC}^{S}$ (MPa) | Specimen Code | $\tau_{u,FC}^{G}$ (MPa) | $\tau_{u,FC}^{G}/\tau_{u,FC}^{S}$ (%) |
|---|---|---|---|---|
| FC3∅_10S | 24.66 | FC3∅_10G | 18.97 | 76.9% |
| FC3∅_12S | 22.24 | FC3∅_12G | 15.03 | 67.6% |
| FC3∅_16S | 18.91 | FC3∅_16G | 7.14 | 37.8% |
| FC4∅_10S | 27.59 | FC4∅_10G | 20.86 | 75.6% |
| FC4∅_12S | 24.39 | FC4∅_12G | 15.62 | 64.0% |
| FC4∅_16S | 21.00 | FC4∅_16G | 10.13 | 48.2% |
| FC5∅_10S | 17.28 | FC5∅_10G | 12.00 | 69.4% |
| FC5∅_12S | 20.52 | FC5∅_12G | 18.33 | 89.3% |
| FC5∅_16S | 17.11 | FC5∅_16G | 6.73 | 39.3% |

$\tau_{u,FC}^{S}$: Average ultimate bond strength of of FC with steel bars, $\tau_{u,FC}^{G}$: Average ultimate bond strength of of FC with GFRP bars.

Concerning the failure pattern of the two types of bars, both of them ended with pull-out failure after recording the ultimate bond strength with the appearance of micro cracks on the surface of the concrete for some specimens, especially those reinforced with conventional steel bars. This indicates the significant dependence of the bond strength on the compressive strength of the concrete at the bond interface [47] due to the high shear stresses generated between the two materials. In Figure 31, the remains of the concrete between the ribs on the bar surface after failure are clearly visible. On the other hand, the bond strength of the GFRP bars is found to be dependent on the resistance of the resin layer that covers the surface of the GFRP bar. This is evident in Figure 32, wherein the rough layer that covered the surface of the bar has peeled off and eroded, followed by the emergence of fine glass fiber filaments randomly around the bonding area. This is due to

friction mechanism between the concrete surface and the high shear stresses between thin layers under the layer of resin covering the surface of the bar [48].

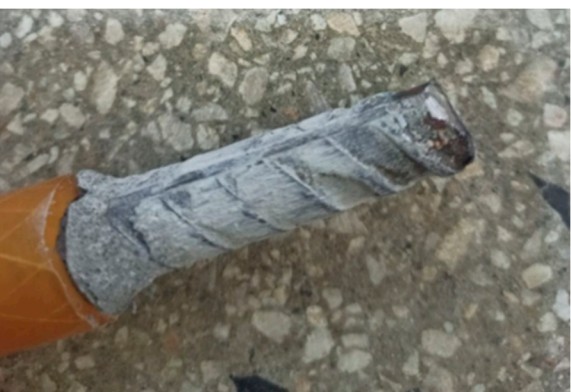

**Figure 31.** Effect of shear stresses generated between bar ribs and concrete at the bond interface.

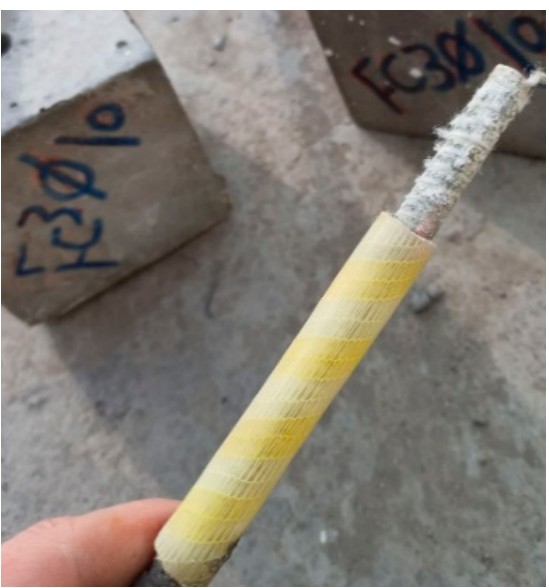

**Figure 32.** Peel off and eroding of the layer covering the GFRP bar during the pull-out test.

### 3.7. Effect of Concrete Type

A comparison of pull-out test results between foamed concrete and normal concrete (Group 1: FC + Steel bar and Group 3: NWC + Steel bar) has been indicated in Table 16. In general, the chemical adhesion between FC and the steel reinforcing bar is superior compared to the NWC and steel bars, regardless of the diameter and the embedded length of the bar. However, the foamed concrete contains stable air voids within it, indicating a weak zone in the specimen. Nonetheless, FC is able to achieve a bond strength equivalent to the normal concrete, which may be attributed to the use of silica fume in the production of foamed concrete. Silica fume has pozzolanic properties which contribute to the hydration process, leading to the formation of calcium hydroxide, water and other forms of calcium silicate hydrate, which improve the bonding between the components of the foamed concrete (fine aggregate and cement paste) and also improve the bond between the concrete and steel reinforcement. Moreover, the use of the superplasticizer enhanced the bond strength of foamed concrete by reducing the w/c ratio. This increases the ability of FC to resist the crushing caused by ribs at the bond interface and impair the bonding area through its effect on bleeding under the reinforcing bar [49]. Additionally, it must be noted that unlike the FC, NWC specimens have shown a sudden shattering and splitting due to

the compressive load applied in an opposite direction to the pull-out load on the machine. Additionally, the cracks in NWC were found to be wider than those of FC specimens, indicating a good cohesion of the foamed concrete components; this is attributed to the use of a low w/c ratio and chemical additives (silica fume) which enhanced the stiffness of FC [50]. Further, a reduction in crack width of FC specimens attracts the interest of building codes that always strive to achieve the same: an increase in the durability of the structure, and restricted corrosion of the reinforcing bars as a result of environmental conditions [51].

**Table 16.** Comparison of average pull-out test results for the two types of concrete (NWC and FC).

| Specimen Code | $\tau_{u,FC}^{S}$ (MPa) | Specimen Code | $\tau_{u,NWC}^{S}$ (MPa) | $\tau_{u,FC}^{S}/\tau_{u,NWC}^{S}$ (%) |
|---|---|---|---|---|
| FC3∅_10S | 24.66 | NWC3∅_10S | 24.83 | 99.3% |
| FC4∅_10S | 22.24 | NWC3∅_12S | 19.35 | 114.9% |
| FC5∅_10S | 18.91 | NWC3∅_16S | 19.17 | 98.6% |
| FC3∅_12S | 27.59 | NWC4∅_10S | 28.65 | 96.3% |
| FC4∅_12S | 24.39 | NWC4∅_12S | 20.57 | 118.6% |
| FC5∅_12S | 21.00 | NWC4∅_16S | 20.5 | 102.4% |
| FC3∅_16S | 17.28 | NWC5∅_10S | 19.11 | 90.4% |
| FC4∅_16S | 20.52 | NWC5∅_12S | 19.45 | 105.5% |
| FC5∅_16S | 17.11 | NWC5∅_16S | 18.32 | 93.4% |

$\tau_{u,FC}^{S}$: Average ultimate bond strength of of FC with steel bars, $\tau_{u,NWC}^{S}$: Average ultimate bond strength of of NWC with steel bars.

The bond strength of most of the FC specimens was observed to be greater than NWC specimens, which may be credited to the presence of hooked-end steel fibers [49]. In addition, the use of hybrid fibers increases the ductility of foamed concrete by increasing the confinement and preventing the expansion of the micro- and macro-cracks.

## 4. Comparison between Experimental and Predicted Bond Strength of FC

### 4.1. FC+ Ribbed Steel Bar

The experimental bond strength for all the fifty-four specimens was computed using the basic equation given by (1). However, prediction of bond strength is dependent on several factors such as bar diameter (∅), concrete cover (c), compressive strength $(f_{c}')$, w/c ratio, density, etc. [49]. Additionally, from the present experimental findings, it has become evident that the bond behavior for shorter bonded lengths (3∅ and 4∅) was different from the bond behavior for the largest bonded length (5∅). The bond strength was found to be inversely proportional to the bar diameter when using shorter bonded lengths, while for the largest bonded length, the ultimate bond strength recorded was for the bar of diameter 12 mm. Therefore, there exists no linear proportionality between the bond strength and the bar diameter of the large bonded lengths.

Orangun et al. [52] proposed an equation for the prediction of bond strength for concrete–steel bars with shorter bonded length (≤4∅), given by Equation (2), in which the bond strength is found to be dependent on bonded length ($l_b$), bar diameter (∅), concrete cover (c), and compressive strength $(f_{c}')$.

$$\tau_{u} = \left[1.22 + 3.23\frac{c}{d_b} + 53\frac{d_b}{l_b}\right]\sqrt{f_{c}'} \tag{2}$$

Kim, Kim, Yun and Lee [50] considered the effect of bonded length ($l_b$), bar diameter (∅) and compressive strength ($f_c$) to be the most influential parameters on the bond strength of steel bars. Equation (3), proposed by Orangun, Jirsa and Breen [52] for the prediction of bond strength, is as follows:

$$\tau_{u} = \left[\frac{37.5}{(∅ + l_b)^{0.25}} - 9.4\right]\sqrt{f_{c}'} \tag{3}$$

Additionally, Zuo and Darwin [51] proposed an Equation (4) for the predicted bond strength of steel bars, where $A_s$ is the area of steel.

$$\tau_u = \left[1.44l_b + (c + 0.5\varnothing) + 56.3A_s\right]\left[0.1\frac{c_m}{c} + 0.9\right]\sqrt[4]{f'_c} \tag{4}$$

Moreover, most researchers agree that the main parameters that affect the concrete-reinforcing bar bonding strength are the bonded length ($l_b$), the diameter of the bar ($\varnothing$) and the concrete cover (c), while the other parameters such as density, compressive strength and w/c ratio were found to be less significant [49].

Therefore, in present study, three parameters, namely bonded length ($l_b$), the diameter of the bar ($\varnothing$) and the concrete cover (c) were adopted as dependent variables, while the density, compressive strength and w/c ratio were considered to be independent variables for developing the prediction equation for FC with steel bars with short bonded lengths ($3\varnothing$ and $4\varnothing$). Meanwhile, for the largest ratio of bonded length ($5\varnothing$), the effect of slip, $S_u$ (refer Table 14), has also been introduced in addition to the previous parameters to predict the bond strength equation. Finally, based on the regression analysis, the predicted Equation (5) for the bond strength was derived as follows:

$$\tau_u = \begin{cases} 2.39\frac{l_b}{\varnothing} - 0.22c + 27.45 & \text{, for } l_b < 5\varnothing \\ 8.6578\frac{S_u\left(\sqrt{l_b+\varnothing}\right)}{c} & \text{, for } l_b \geq 5\varnothing \end{cases} \tag{5}$$

Figures 33 and 34 compare the ratios of average experimental ultimate bond strength to the average predicted ultimate bond strength of the proposed Equation (5), and the previously derived equations existing in the literature [50]. All results of the comparison are displayed in Table 17.

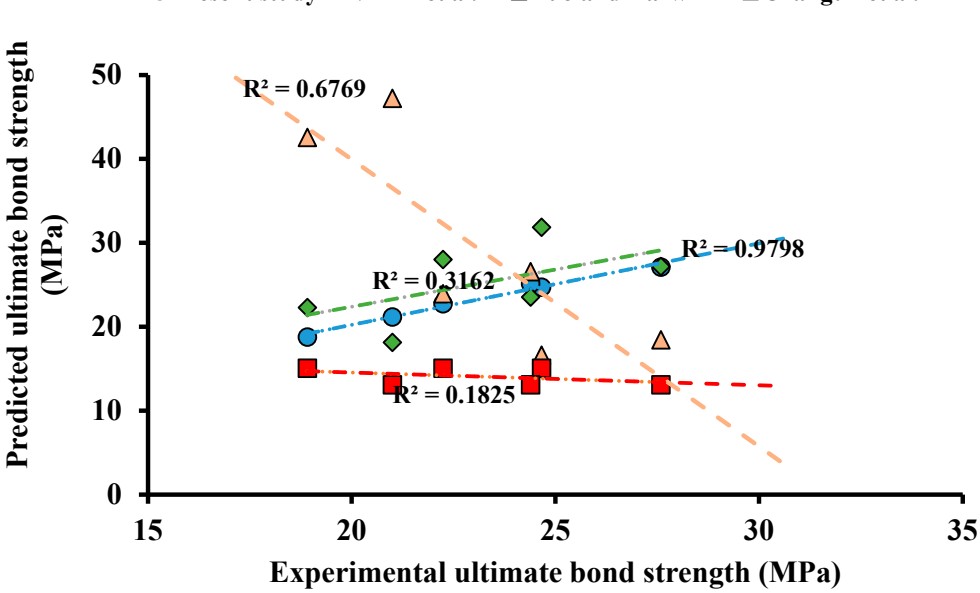

**Figure 33.** Comparison between the experimental and predicted ultimate bond strength for bonded lengths $3\varnothing$ and $4\varnothing$ [50,52].

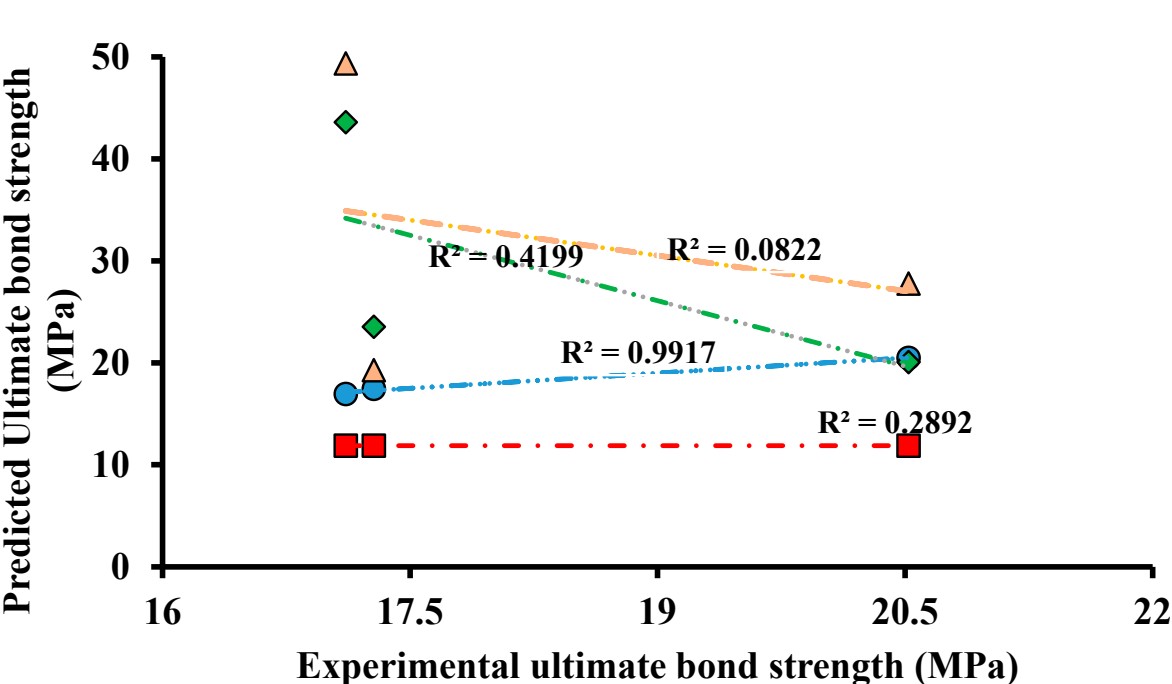

**Figure 34.** Comparison between the experimental and predicted ultimate bond strength for bonded length 5∅ [50,52].

**Table 17.** A comparison between the experimental and the predicted results of average ultimate bond strength (MPa) for FC + steel bar.

| Specimen Code | (MPa) | | | | |
|---|---|---|---|---|---|
| | $\tau_{u_{exp}}$ [Present Study] | $\tau_{u_{pred}}$ [Present Study] | $\tau_u$ [30] | $\tau_u$ [31] | $\tau_u$ [32] |
| FC3∅_10S | 24.7 | 24.5 | 31.9 | 16.6 | 15.1 |
| FC4∅_10S | 27.6 | 27.0 | 27.2 | 18.4 | 13.1 |
| FC5∅_10S | 20.5 | 20.5 | 23.5 | 19.3 | 11.9 |
| FC3∅_12S | 22.2 | 22.3 | 28.0 | 24.0 | 15.1 |
| FC4∅_12S | 24.4 | 25.0 | 23.5 | 26.6 | 13.1 |
| FC5∅_12S | 17.3 | 17.5 | 20.1 | 27.8 | 11.9 |
| FC3∅_16S | 19.0 | 18.4 | 22.3 | 42.6 | 15.1 |
| FC4∅_16S | 21.0 | 21.5 | 18.1 | 47.2 | 13.1 |
| FC5∅_16S | 17.1 | 17.0 | 43.6 | 49.4 | 11.9 |

*4.2. FC+ Sand-Coated GFRP Bar*

Few equations have been given in the literature for predicting the bond strength of concrete–GFRP bars. ACI [53] gave the following Equation (6) for the prediction of bond strength:

$$\tau_u = \left[0.332 + 0.025\frac{c}{\varnothing} + 8.3\frac{\varnothing}{l_b}\right]\sqrt{f'_c} \tag{6}$$

where '$l_b$' is the bonded length, '$\varnothing$' is the diameter of the bar and '$c$' is the concrete cover.

Additionally, according to Eligehausen et al. [54], relation for bond strength is given by Equation (7):

$$\frac{\tau}{\tau_1} = \left[\frac{S}{S_1}\right]^{\alpha} \tag{7}$$

where '$S$' is the slip at failure and '$\alpha$' is a curve fitting parameter $\leq 1$.

In the case of GFRP bars, the smallest diameter, 10 mm, was not affected by the concrete cover, and was completely pulled out during the test without leaving any crack on the surface of concrete specimens. Therefore, the bond strength values were considered to be governed by the bonded length and the bar's diameter when performing the linear regression analysis. The resulting equation was found to have a high correlation of 0.86; therefore, to ensure the reliability of the results, the findings were compared with the results obtained from the equation existing in the literature [54], as shown in Figure 35. Meanwhile, for the larger diameters, 12 mm and 16 mm, in addition to the diameter and bonded length of GFRP bar, bond strength was observed to be affected by the concrete cover (c) and slip at failure ($S_u$). This is because the cover provided partial confinement to the bars, leading to the partial pulling off of the bar during the pull-out test. Therefore, based on the regression analysis and curve fitting value $\alpha = 0.4$ (refer Equation (7)), the predicted Equation (8) was derived as follows:

$$\tau_u = \begin{cases} -0.4798\frac{l_b}{\varnothing} + 10.821 & , \text{ for } \varnothing = 10 \text{ mm} \\ -13\frac{c}{S_u\left[\sqrt[4]{l_b+\varnothing}\right]} + 25.656 & , \text{ for } \varnothing > 10 \text{ mm} \end{cases} \tag{8}$$

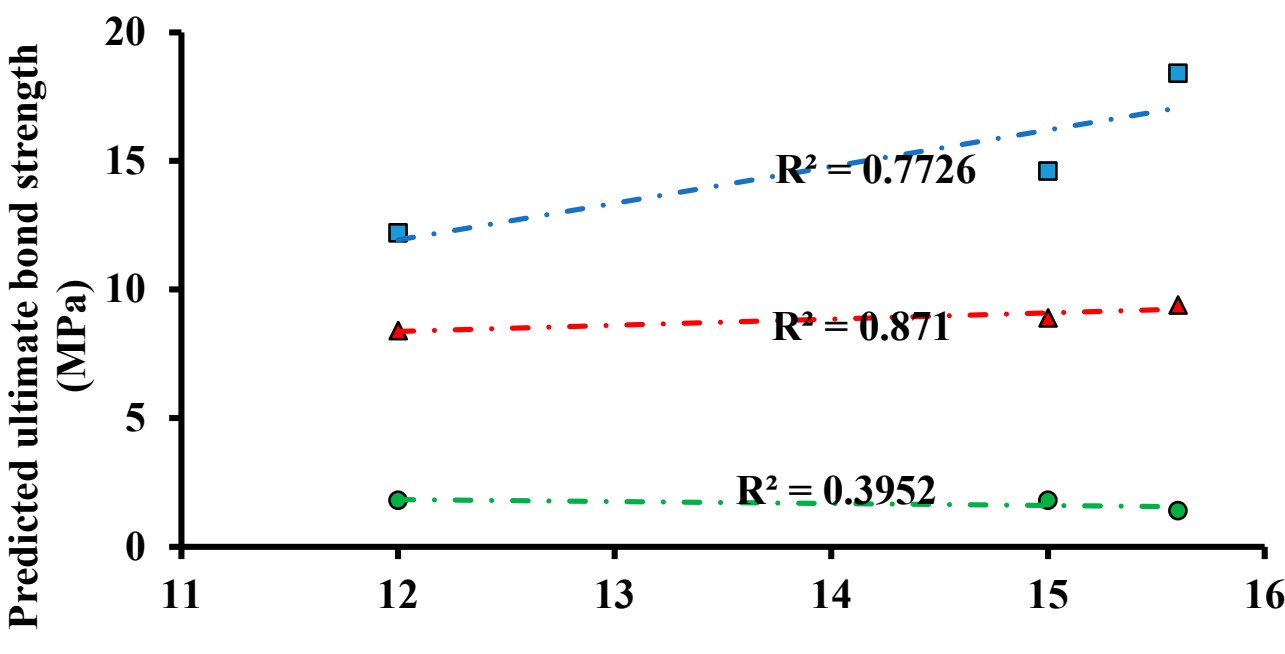

**Figure 35.** Comparison between the experimental and predicted ultimate bond strength for the 10 mm diameter GFRP bars [54].

From Figure 36, it has been observed that experimental results were closely related to the predicted bond strength ($R^2 = 0.9289$). All results of the comparison are presented in Table 18.

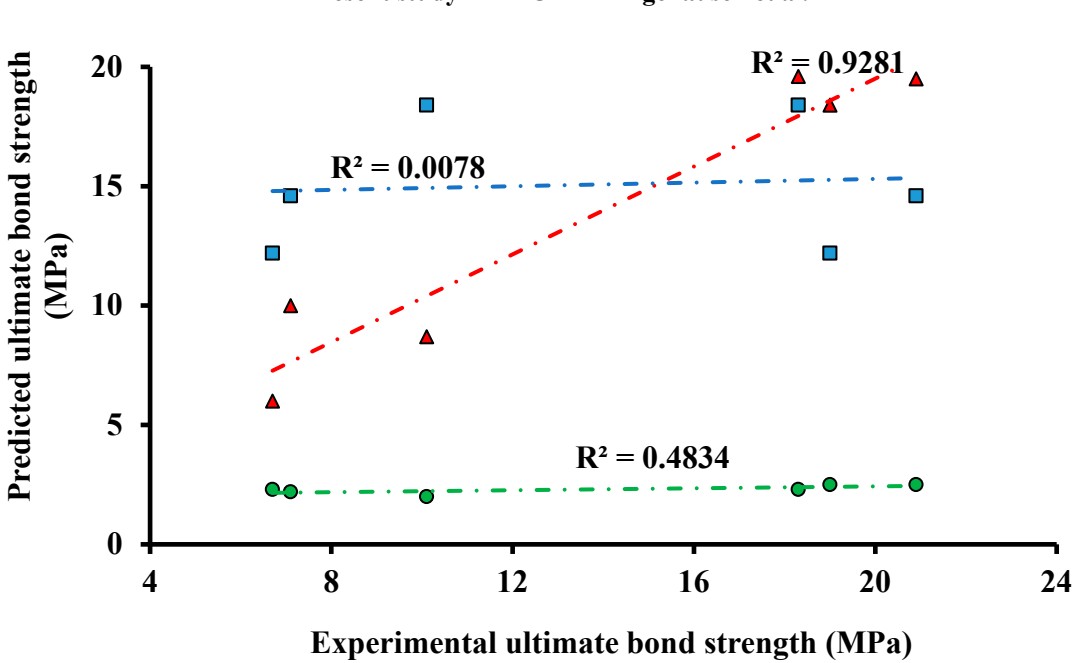

**Figure 36.** Comparison between the experimental and predicted ultimate bond strength for the 12 mm and 16 mm diameter GFRP bars [54].

**Table 18.** Comparison between the experimental and the predicted results of average ultimate bond strength (MPa) for FC + GFRP.

| Specimen Code | (MPa) | | | |
|---|---|---|---|---|
| | $\tau_{u_{exp}}$ (Present Study) | $\tau_{u_{pred}}$ (Present Study) | $\tau_u$ (ACI [53]) | $\tau_u$ (Eligehausen et al. [54]) |
| FC3∅_10G | 19.0 | 18.4 | 18.4 | 1.4 |
| FC4∅_10G | 20.9 | 19.5 | 14.6 | 1.8 |
| FC5∅_10G | 18.3 | 19.6 | 12.2 | 1.8 |
| FC3∅_12G | 15.0 | 18.0 | 18.4 | 2.3 |
| FC4∅_12G | 15.62 | 9.5 | 14.6 | 2.5 |
| FC5∅_12G | 12 | 8.4 | 12.2 | 2.5 |
| FC3∅_16G | 7.14 | 10.0 | 18.4 | 2 |
| FC4∅_16G | 10.3 | 8.7 | 14.6 | 2.2 |
| FC5∅_16G | 6.7 | 6.0 | 12.2 | 2.3 |

## 5. Conclusions

The present study aims to investigate the bond strength of foamed concrete (FC) and normal weight concrete (NWC) with ribbed steel and sand-coated GFRP bars through a direct pull-out test. The test was conducted on fifty-four cube specimens, considering different variables, viz., the type of reinforcement (sand-coated glass fiber-reinforced polymer (GFRP) and ribbed steel bars), the diameter of the reinforcing bars (10 mm, 12 mm, and 16 mm) and the bonded length ratio (3∅, 4∅, and 5∅). The following are the major conclusions obtained from the study:

1.  The use of hybrid fibers both short and long (Hs and Pp fibers) significantly improved the mechanical properties of the foam concrete (FC) and reduced crack formation and propagation. For shorter bonded lengths of steel and GFRP bars, the bond strength was found to obey an inverse relation, while for longer bonded lengths, no proportionality was observed. On the other hand, for a constant diameter of bar, the bond strength was found to be maximum for a bonded length of 4∅ in both steel and GFRP bars.

2. For the steel bar of diameter 10 mm, the bond strength of FC4∅_10S was found to be 11.9% and 18.7% higher than FC3∅_10S and FC5∅_10S, respectively. Similarly, for diameter 12 mm, the bond strength of FC4∅_12S was found to be 9.7% and 59.7% greater compared to FC3∅_12S and FC54∅_12S, respectively. Additionally, for a 16 mm diameter bar, the bond strength of FC4∅_16S with a bonded length of 64 mm was found to be 11.1% and 22.7% higher than FC3∅_16S and FC5∅_16S, respectively.

3. For the GFRP bar of diameter 10 mm, the bond strength of FC4∅_10G was found to be 10.0% and 30.2% higher than FC3∅_10G and FC5∅_10G, respectively. For the bar of diameter 12 mm, the bond strength of FC4∅_12G was found to be 4.0% greater compared to FC3∅_12G, and 13.8% lesser compared to FC5∅_12G. Additionally, for the bar of diameter 16 mm, the bond strength of FC4∅_16G was found to be 41.9% and 50.5% higher than FC3∅_16G and FC5∅_16G, respectively.

4. In case of GFRP bars, the bond strength was enhanced by 165.7% and 105.9%, corresponding to the bonded length 3∅ and 4∅, respectively, upon reducing the diameter from 16 mm to 10 mm. On the other hand, for a longer bonded length (5∅), the bond strength was found to be maximum for the 12 mm diameter bar.

5. Through the regression analysis, equations of very high correlation were predicted; these represent the practical results of the bond strength of FC with steel bars of short (3∅, 4∅) and long (5∅) bonded lengths. The predicted bond strengths showed a good correlation with the experimental values, and were also found to be comparable with the results obtained using equations existing in the literature.

**Author Contributions:** Conceptualization, S.M.A., R.H., S.S., H.M.N. and M.M.S.S.; Methodology, S.M.A., R.H. and S.A.; Software, S.A., H.M.N. and M.M.S.S.; Formal analysis, S.S. and H.M.N.; Investigation, R.H.; Resources, S.S.; Data curation, S.S.; Writing—original draft, S.M.A., R.H. and S.A.; Writing—review & editing, S.M.A., S.S., H.M.N. and M.M.S.S.; Visualization, H.M.N. and M.M.S.S.; Supervision, M.M.S.S.; Project administration and S.A.; Funding acquisition, M.M.S.S. All authors have read and agreed to the published version of the manuscript.

**Funding:** This research was funded by the Ministry of Science and Higher Education of the Russian Federation within the framework of the state assignment No. 075-03-2022-010 dated 14 January 2022 and No. 075-01568-23-04 dated 28 March 2023(Additional agreement 075-03-2022-010/10 dated 09 November 2022, Additional agreement 075-03-2023-004/4 dated 22 May 2023), FSEG-2022-0010.

**Institutional Review Board Statement:** Not applicable.

**Informed Consent Statement:** Not applicable.

**Data Availability Statement:** Not applicable.

**Acknowledgments:** The authors extend their thanks to the Ministry of Science and Higher Education of the Russian Federation for funding this work.

**Conflicts of Interest:** There is no conflict of interest.

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
