# Peer review of "Effect of Using Glass Fiber Reinforced Polymer (GFRP) and Deformed Steel Bars on the Bonding Behavior of Lightweight Foamed Concrete"

_buildings, doi:10.3390/buildings13051153_

Round 1

Reviewer 1 Report

The authors should put a tremendous effort to do the required comments. My comments are listed as follows:
- The whole abstract was not properly structured, too long and more results should be added.
-The introduction must be much improved and a recent researches should be added.
- There are several studies reporting similar data but authors offer no comparison with these results.
- Meaningful conclusions are needed as conclusions are general and is common sense. Conclusion must be reconstructed & simplified in 4-5 main important points with supporting results.
- The quality of the all figures is very poor and should be improved.
- The English grammar need careful attention and correction.

Author Response

Dear Reviewer,

On behalf of all authors, thanks for all the effort and time you put into reviewing our manuscript. Where the comments were useful and contributed effectively to raising the quality of the manuscript.

Kind regards,

Reviewer 2 Report

Authors have worked on "Effect of Using Glass Fiber Reinforced Polymer (GFRP) and Deformed Steel Bars on the Bonding Behavior of Lightweight Foamed Concrete". Excellent work from the authors especially in concern with the natural resources, exploitation, and performance related to foam concrete. The manuscript can be accepted for publication after attending to the below-mentioned minor comments.

1. Authors have missed citing the relevant references wherever the bold statements are used. 

2. The work is limited to the use of glass fiber-reinforced polymer. Suggested incorporating the work carried out by the researchers by considering other types of fibers. For example...... Please refer to the last comment. So that it will gel with the flow of research.

3. Suggested to incorporate the overall impact of the study in the conclusion and abstract.

4. Highlight and differentiate the types of sand used in Figure 1. It is incomplete.

5. Compare the chemical composition of both the binders used. SF and OPC.

6. Crosscheck Table 11.

7. Highlight the overall methodology followed for this research using a simple flowchart.

8. Discuss the results with available literature.

9. Incorporate at least 5-8 references from BUILDINGS.

10. Follow the below-mentioned references and incorporate them wherever it is necessary.

A) Manjunatha, M., Seth, D., & Balaji, K. V. G. D. (2021). Role of engineered fibers on fresh and mechanical properties of concrete prepared with GGBS and PVC waste powder–An experimental study. Materials Today: Proceedings47, 3683-3693.

B) Reshma, T. V., Manjunatha, M., Bharath, A., Tangadagi, R. B., Vengala, J., & Manjunatha, L. R. (2021). Influence of ZnO and TiO2 on mechanical and durability properties of concrete prepared with and without polypropylene fibers. Materialia18, 101138.

Author Response

On behalf of all authors, thanks for all the effort and time you put into reviewing our manuscript. Where the comments were useful and contributed effectively to raising the quality of the manuscript.

Kind regards,

Reviewer 3 Report

The paper needs following revisions:

Introduction should contain novelty of the study

Add details for test specimens

Most importantly is that:

The figures can combine inorder to reduce the number of pages. The size of figures is too big. Therefore it is hard to read it.

Author Response

(The authors gave the same response as above.)
